# HELP: Hardware-Adaptive Efficient Latency Prediction for NAS via Meta-Learning

**Hayeon Lee**[1*] **Sewoong Lee**[1*] **Song Chong**[1] **Sung Ju Hwang**[1,2]
KAIST[1], AITRICS[2], Seoul, South Korea
{hayeon926, dltpdnd21, songchong, sjhwang82}@kaist.ac.kr

## Abstract

For deployment, neural architecture search should be hardware-aware, in order to satisfy the device-specific constraints (e.g., memory usage, latency and energy consumption) and enhance the model efficiency. Existing methods on hardware-aware NAS collect a large number of samples (e.g., accuracy and latency) from a target device, either builds a lookup table or a latency estimator. However, such approach is impractical in real-world scenarios as there exist numerous devices with different hardware specifications, and collecting samples from such a large number of devices will require prohibitive computational and monetary cost. To overcome such limitations, we propose Hardware-adaptive Efficient Latency Predictor (HELP), which formulates the device-specific latency estimation problem as a meta-learning problem, such that we can estimate the latency of a model's performance for a given task on an unseen device with a few samples. To this end, we introduce novel hardware embeddings to embed any devices considering them as black-box functions that output latencies, and meta-learn the hardware-adaptive latency predictor in a device-dependent manner, using the hardware embeddings. We validate the proposed HELP for its latency estimation performance on unseen platforms, on which it achieves high estimation performance with as few as 10 measurement samples, outperforming all relevant baselines. We also validate end-to-end NAS frameworks using HELP against ones without it, and show that it largely reduces the total time cost of the base NAS method, in latency-constrained settings. Code is available at https://github.com/HayeonLee/HELP.

## 1 Introduction

Neural Architecture Search (NAS) [39, 1, 25, 22, 21, 36, 5], which aims to search for the optimal architecture for a given task, has achieved a huge practical success by overcoming the sub-optimality of manual architecture designs. However, for NAS to be truly practical in real-world scenarios, it should be hardware-aware. That is, we need to search for the neural architectures that satisfy diverse device constraints (e.g., memory footprint, inference latency, and energy consumption). Due to the practical importance of the problem, many existing works propose to take into account the hardware efficiency constraints (mostly latency) in the search process [29, 3, 34, 32, 6, 2, 37, 4, 33].

However, one of the main challenges with such hardware-aware NAS is that collecting training samples (e.g., architecture-latency pairs on each target device) to build reliable prediction models for the efficiency metrics, is computationally costly or requires the knowledge of the hardware devices. Existing hardware-aware NAS methods usually require a large number of samples (e.g., 5k) and train metric predictors for each device from scratch. Additionally, most works [29, 3, 34, 32, 6] are task-specific, and thus such collection of performance samples should be done from scratch, for a new task. Thus, the sample collection process is prohibitively costly when we consider real-world

---

*These authors contributed equally to this work.

35th Conference on Neural Information Processing Systems (NeurIPS 2021).

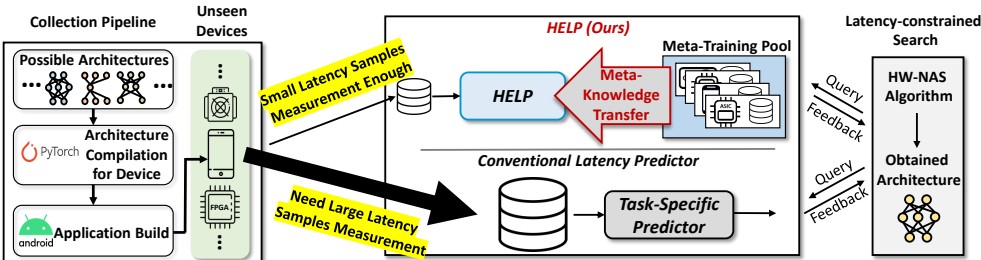

Figure 1: **Concept.** Conventional latency estimation methods require a large number of architecture-latency pairs to build a prediction model separately for each device, which is inefficient. Contrarily, the proposed HELP uses a single meta-latency predictor that can fast adapt to any unknown device by collecting only a few latency measurements from it, by utilizing the meta-knowledge of the source device pool.

deployment of a model to a large number of hardware devices, for any tasks. OFA [4] alleviate the search cost by utilizing a high-performance network trained on ImageNet as the supernet, which does not require training the models from scratch. Yet, they are still sub-optimal since building device-specific predictors for metric still requires a large number of samples to be collected, to build a layer-wise latency predictor for each device. This could take multiple hours depending on the task and the device, and becomes a bottleneck for latency-constrained NAS. BigNAS [38] considers FLOP as the efficiency metric, but this is a highly inaccurate proxy since latency of an architecture could differ based on its degree of parallelism and memory access cost, for the same FLOP.

To overcome such limitations, we propose a novel sample-efficient latency predictor, namely Hardware-adaptive Efficient Latency Predictor (HELP), which supports the latency estimation for multiple devices with a **single** model, by allowing it to rapidly adapt to unseen devices with only a few latency measurements collected from each device. Specifically, we formulate the latency prediction problem as a few-shot regression task of estimating the latency given an architecture-device pair, and propose a novel hardware embedding that can embed any devices by utilizing the latencies of the reference architectures on each device as its embeddings. Then, we propose a meta-learning framework which combines amortized meta-learning with gradient-based meta-learning, to learn the latency predictor to generalize across multiple devices, utilizing the proposed hardware embeddings. This allows the model to transfer knowledge learned from known devices to a new device, and thus to achieve high sample efficiency when estimating the latency of unseen devices.

HELP is highly versatile and general, as it is applicable to any hardware platforms and architecture search spaces, thanks to our device-agnostic hardware embeddings. Also, HELP can be coupled with any NAS frameworks to reduce its computational bottleneck in obtaining latency-constrained architectures. Especially, when coupled with rapid NAS methods such as MetaD2A [16], OFA [4] and HAT [33], HELP can perform the entire latency-constrained NAS process for a new device almost instantly. We validate the latency estimation performance of HELP on the NAS-Bench-201 space [9] with various devices from different hardware platforms, utilizing the latency dataset for an extensive device pool in HW-NAS-Bench dataset [18]. The results show that our meta-learned predictor successfully generalize to unseen target devices, largely outperforming baseline latency estimation methods using at least $90\times$ less measurements. Then, we combine HELP with existing NAS methods [4, 33] and show that our meta-latency predictor largely reduce their total search cost. To summarize, the main contributions of this paper are as follows:

- We formulate the latency estimation of a neural architecture for a given device as a few-shot regression problem, which outputs the latency given an architecture-device pair as the input.

- To represent heterogeneous hardware devices, we propose a novel device-agnostic hardware embedding, which embeds a device by its latencies on reference architectures.

- We propose a novel latency predictor, HELP, which meta-learns a few-shot regression model to generalize across hardware devices, that can estimate the latency of an unseen device using only few measurements. HELP obtains significantly higher latency estimation performance over baselines with minimal total time cost.

- We further combine HELP with existing NAS frameworks to show that it leads to find latency-constrained architectures extremely fast, eliminating the latency-estimation bottleneck in the hardware-constrained NAS.

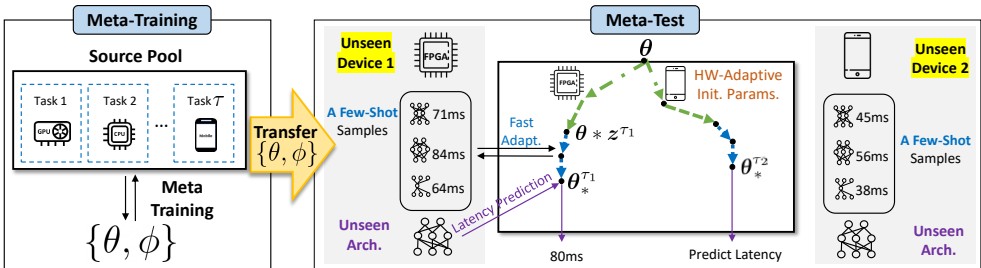

Figure 2: **Overview.** For the hardware-adaptive latency estimation of an unseen device for latency-constrained NAS, we introduce a latency-based hardware embedding and a $z$ modulator of the initial parameters. By formulating the sample-efficient NAS problem as a few-shot regression problem under the meta-learning framework, our meta-learned predictor successfully exploits meta-knowledge $\theta$ from the source device pool, to achieve high sample efficiency on unseen devices.

## 2 Related Work

**Latency Prediction in NAS**   Hardware-aware NAS [29, 3, 4, 34, 32, 13, 33, 26] aims to design neural architectures that achieves a good trade-off between accuracy and latency for efficient deployment to target devices. While they need to consider the actual latencies of the architectures on a target device, evaluating them while searching for architectures is costly. Thus, most existing works replace it with proxies, such as FLOPs [38], but they are inaccurate measure of latencies. Earlier hardware-aware NAS methods have been used a layer-wise latency predictor (lookup table) [29, 3, 4, 34] which sums up the latencies measured for each operation in the given neural networks. While this is better than FLOPs, they do not accurately capture the complexities of multiple layer execution on real hardware devices. Thus, recent methods use an end-to-end latency predictor [10, 33] that is trained with the latency measurements from the target device, to improve the latency prediction accuracy. BRP-NAS [10], which is a GCN-based model, is currently the state-of-the-art end-to-end latency predictor. However, this method is also limited in that it requires a large number of architecture-latency pairs for each device. Since the latency estimator cannot generalize to a new device, whenever a new device is given, the NAS system needs to build a new latency estimator, which may take hours to finish. The proposed method significantly reduces the total building cost of the latency predictor for each device, by using a single meta-learned predictor and adapting it to a new device with only a few measurements from it.

**Meta-learning and Meta-NAS**   Meta-learning (learning to learn) [30] aims to learn a model that generalizes over a distribution of tasks rather than a single task, such that the model meta-learned over a large number of tasks rapidly adapts to an unseen task. The performance of existing methods [31, 28, 12, 24, 17] is usually evaluated on few-shot classification tasks, where the model classifies between instances of unseen classes given only a few training examples. Recently, several works have proposed to utilize meta-learning for NAS. Most of them focus on few-shot classification tasks to search for the architectures and parameters that can generalize well to a new task [11, 20, 27] with gradient-based meta-learning. However, they have limited practical applicability since the computational cost of meta-learning is prohibitively large, for NAS under a standard many-shot setting. A recently proposed meta-NAS framework with amortized meta-learning, MetaD2A [16], which utilizes a set encoder to encode a task and uses a graph decoder to generate a task-adaptive architecture, obtained state-of-the-art performance on unseen datasets, with minimal search cost. We also propose a similar amortized meta-learning framework, for hardware-adaptive NAS, based on a novel hardware device embedding. However, after obtaining the device-conditioned initialization parameters, we perform further inner gradient steps for device-specific adaptation, unlike MetaD2A [16].

## 3 Method

Our goal is to design a prediction model that can accurately predict the efficiency metric for a novel architecture-device pair, using a small number of performance samples from the device. While our method is generally applicable to any efficiency metrics that can be measured from the device (e.g. latency, energy consumption, and memory footprint) we focus on the latency prediction in this work.

### 3.1 Problem Definition

Assume that we are given a task specification $\tau = \{h^\tau, \mathbf{X}^\tau, \mathbf{Y}^\tau\}$ where $h^\tau \in \mathcal{H}$ is a hardware device, $\mathbf{X}^\tau \subset \mathcal{X}$ is a set of neural architectures, and $\mathbf{Y}^\tau \subset \mathcal{Y}$ is a set of latencies of $\mathbf{X}^\tau$ directly measured on the hardware device $h^\tau$. Then, our goal is to learn a regression model $f(x; \boldsymbol{\theta}) : \mathcal{X} \rightarrow \mathbb{R}$ parameterized by $\boldsymbol{\theta}$ that estimates the latency $y \in \mathbf{Y}^\tau$ of a neural architecture $x \in \mathbf{X}^\tau$ for a given hardware device $h^\tau$ by minimizing empirical loss $\mathcal{L}$ (e.g. mean squared error) on the predicted values $f(\mathbf{X}^\tau; \boldsymbol{\theta})$ and actual measurements $\mathbf{Y}^\tau$ as follows:

$$\min_{\boldsymbol{\theta}} \mathcal{L}\big(f^\tau(\mathbf{X}^\tau; \boldsymbol{\theta}), \mathbf{Y}^\tau\big) \tag{1}$$

Learning such a regression model seems like a simple problem, since we can collect any number of measurements from any devices. However, this is a more challenging problem than it seems:

1. Since we cannot generalize across devices, we need to learn $N$ predictors $\{f^\tau(\cdot; \boldsymbol{\theta}^\tau)\}_{\tau=1}^N$ for $N$ devices, collecting performance samples and training the performance predictor separately for each device, which requires prohibitive computational costs with large number of devices.

2. Even when assuming that we learn a device-specific predictor, in order not to overfit the regression model, we need to collect *a large number of* architecture-latency sample pairs for each device (e.g. 2k [33], 5k [23] samples) to achieve reliable prediction performance.

3. With lack of generalization ability across devices and architectures, the NAS framework needs to repeat the time-consuming sample collection process whenever a new device is given, that may take hours to complete. This will become a computational bottleneck even for a rapid meta-NAS framework such as MetaD2A [16].

To overcome such limitations, we propose a **single** predictor $f(\cdot; \boldsymbol{\theta})$ which can generalize across devices and architectures, by **fast** adapting to a new target device and architecture that are unseen during training, by collecting only a **small number** of architecture-latency pairs from the device ($X^\tau \ll \mathbf{X}^\tau, Y^\tau \ll \mathbf{Y}^\tau$). We achieve this goal with a meta-learning framework that can transfer knowledge obtained from the device and architecture pool $p(\tau)$.

### 3.2 Hardware-adaptive Latency Prediction with Device Embeddings

While the measured latency $y \leftarrow (x, h)$ is dependent on both the device type $h$ and the architecture $x$, existing latency prediction models takes the form of $f(x; \boldsymbol{\theta})$, ignoring the device-specific constraints, since the latency predictor is learned for each device separately. This results in poor performance when learning a single latency predictor to perform metric estimation on multiple devices, including unseen ones, which is our main objective. Thus, we propose hardware-conditioned prediction model:

$$f(x, h; \boldsymbol{\theta}) : \mathcal{X} \times \mathcal{H} \rightarrow \mathbb{R} \tag{2}$$

that can predict the latency differently depending on the device type $h$, even for the same architecture $x$. A crucial challenge of our hardware-conditioned prediction model is how to represent the hardware device $h$, for all devices regardless of their platform types. This is not a trivial problem since the physical architecture of hardware devices could be very different (e.g. CPU vs FPGA). Thus, we simply consider the device as a black box function which outputs the inference latency given an architecture, instead of directly modeling the hardware devices. Then, we obtain the latencies of the device on a fixed set of reference neural architectures as follows:

$$V_h = \{y_1^*(x_1, h), y_2^*(x_2, h), ..., y_d^*(x_d, h)\} \tag{3}$$

where $\mathcal{E}$ is the set of the reference neural architectures $\{x_1, x_2, ..., x_d\} \subset \mathcal{X}$, fixed across all tasks for both meta-training and meta-test, and $d$ is the number of the reference architectures; in our experiments, we set $d = 10$. Further, $y_i^*(x_i, h) = \{y_i(x_i, h) - min(V_h^{(0)})\}/\{max(V_h^{(0)}) - min(V_h^{(0)})\}$ are standardized latency values ranging from 0 to 1, where $V_h^{(0)} = \{y_1(x_1, h), y_2(x_2, h), ..., y_d(x_d, h)\}$. Since this set of reference devices should be representative, we select them to be diverse and heterogeneous. As for the reference architectures, we randomly sample them from the search space. For more detailed descriptions of the reference devices and architectures, please see the supplementary file. The proposed black-box treatment of hardware devices, and the latency-based hardware embedding allows us to embed a new device without considering its detailed hardware specification.

### 3.3 Meta-Learning the Hardware-adaptive Latency Predictor

To tackle the few-shot regression problem for multiple devices by utilizing the collected pool of devices and architectures $p(\tau)$, we propose a novel hardware-adaptive meta-learning framework of the latency predictor that meta-learns $f(x, h; \boldsymbol{\theta})$ across a task distribution $p(\tau)$ to rapidly adapt the predictor $f(x, h; \boldsymbol{\theta}^\tau)$ to an unseen neural architecture $x$ given the task specification $\tau = \{h^\tau, V_h, \mathbf{X}^\tau, \mathbf{Y}^\tau\}$. During the meta-training phase, we leverage the episodic training strategy, in which we simulate a few-shot regression task for each iteration by randomly sampling task $\tau$ from the device-architecture pool $p(\tau)$ and splitting $\tau$ as training set $\mathcal{D} = \{h^\tau, X^\tau, Y^\tau\}$ and test set $\tilde{\mathcal{D}} = \{h^\tau, \tilde{X}^\tau, \tilde{Y}^\tau\}$ where $X^\tau \subset \mathbf{X}$ and $\tilde{X}^\tau \subset \mathbf{X}$ denote sets of neural architecture samples, $X^\tau$ is the set of few-shot samples $|X^\tau| \ll |\mathbf{X}|$. Note that there is no overlap between them; that is, $X^\tau \cap \tilde{X}^\tau = \emptyset$. $Y^\tau \subset \mathbf{Y}^\tau$ and $\tilde{Y}^\tau \subset \mathbf{Y}^\tau$ denote the sets of corresponding latency values of neural architectures $X^\tau$ and $\tilde{X}^\tau$, measured on device $h^\tau$, respectively. Basically, for each task $\tau$, we obtain the hardware-adaptive prediction model $f(X, V_h^\tau; \boldsymbol{\theta}^\tau)$ as a function of $V_h^\tau$. Formally, we meta-train the latency predictor to minimize the test loss $\mathcal{L}(\cdot; \hat{\mathcal{D}}^\tau)$ by optimizing the following task-adaptive meta-learning objective:

$$\min_{\boldsymbol{\theta}} \sum_{\tau \sim p(\tau)} \mathcal{L}\big(f\big(\tilde{X}^\tau, V_h^\tau; \boldsymbol{\theta}^\tau\big), \tilde{Y}^\tau\big) \tag{4}$$

We can simply use the task embedding $V_h^\tau$ to obtain a task-conditioned latency predictor, in which case we are using an amortized meta-learning framework similar to one proposed in Lee et. al. [16], which aims to meta-learn a dataset-adaptive performance predictor and a NAS framework. However, we can further perform few-shot adaptation with the few latency measurements collected from the target device, by conducting inner gradient updates with them as follows:

$$\boldsymbol{\theta}_{(t+1)}^\tau = \boldsymbol{\theta}_{(t)}^\tau - \alpha \nabla_{\boldsymbol{\theta}_{(t)}} \mathcal{L}(f(X^\tau, V_h^\tau; \boldsymbol{\theta}_{(t)}), Y^\tau) \quad \text{for } t = 1, \ldots, T \tag{5}$$

where $t$ denotes the $t_{th}$ inner gradient step, $T$ is the total number of inner gradient steps, and $\alpha$ denotes the multi-dimensional global learning rate vector [19]. This meta-learning formulation allows us to adapt to a new device rapidly, by using the knowledge of the devices used for meta-training. However, when we are encountered with a completely new device that has little relatedness to any devices from the meta-device pool, it will be helpful to deviate more from the meta-knowledge captured by $\theta_{(0)}^\tau$ [15]. To this end, we introduce a hardware-adaptive modulator $\boldsymbol{z}^\tau = g(V_h^\tau; \boldsymbol{\phi}) : \mathbb{R}^d \to \mathbb{R}^{d_{\boldsymbol{\theta}}}$ parameterized by $\phi$ to modulate the initial parameter as $\boldsymbol{\theta}_{(0)} = \boldsymbol{\theta} * \boldsymbol{z}^\tau$, where $\boldsymbol{\theta}_{(0)}$ is the new initialization for hardware $h^\tau$. Following [15], we set $\boldsymbol{\theta}_{(0)} \leftarrow \boldsymbol{\theta} \circ \boldsymbol{z}^\tau$ for weights and $\boldsymbol{\theta}_{(0)} \leftarrow \boldsymbol{\theta} + \boldsymbol{z}^\tau$ for biases with an element-wise multiplication operator $\circ$. This leads to the following update rule:

$$\boldsymbol{\theta}_{(0)}^\tau = \boldsymbol{\theta} * \boldsymbol{z}^\tau \tag{6}$$

$$\boldsymbol{\theta}_{(t+1)}^\tau = \boldsymbol{\theta}_{(t)}^\tau - \alpha \nabla_{\boldsymbol{\theta}_{(t)}^\tau} \mathcal{L}(f(X^\tau, V_h^\tau; \boldsymbol{\theta}_{(t)}^\tau), Y^\tau) \quad \text{for } t = 1, \ldots, T \tag{7}$$

where $T$ is the number of inner gradient steps. Then, the final meta-learning objective is as follows:

$$\min_{\boldsymbol{\theta}, \boldsymbol{\phi}} \sum_{\tau \sim p(\tau)} \mathcal{L}\big(f\big(\tilde{X}^\tau, V_h^\tau; \boldsymbol{\theta}_{(T+1)}^\tau\big), \tilde{Y}^\tau\big) \tag{8}$$

Thus, we meta-learn both the model parameters for the hardware-adaptive latency predictor, and the modulator for the shared initial parameters.

**Few-shot Adaptation to Unseen Devices (Meta-Test)** We now describe how to use our meta-learned latency prediction model $f(\cdot; \boldsymbol{\theta})$ to estimate the latency of an architecture on a novel device $h^\upsilon$ that is unseen during meta-training. The task-specific predictors [33, 4, 23] need a large amount of latency measurements from an unseen device $h^\upsilon$, over diverse architectures in order not to overfit, which may take an excessive time to collect. However, our model is able to measure the latency values $y^\upsilon$ of a new architecture $\tilde{x}^\upsilon$ by collecting only few latency measurements from it (we use 10 or 20), using the device-conditioned meta-learning. Given a latency prediction task of an architecture $\tilde{x}^\upsilon$ on a novel device $h^\upsilon$, $\upsilon = \{h^\upsilon, X^\upsilon\}$, we first obtain its hardware device embedding $V_h$ by obtaining its latencies on a fixed set of reference architectures, following Equation (3). Then we use the device embedding $V_h$ to obtain the device-optimized parameters of the latency predictor $\boldsymbol{\theta}_{(T+1)}^\upsilon$, following Equation (6) and (7). Then, we use the device-optimized latency predictor $f(\cdot, V_{h^u}; \boldsymbol{\theta}_{(T+1)}^\upsilon)$ to measure the latency of $\tilde{x}^\upsilon$. We can further combine this meta-latency predictor with a NAS method to perform latency-constrained NAS for a novel device.

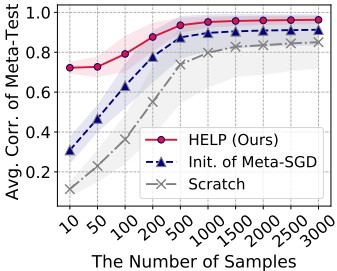

Figure 3: Latency estimation performance as a function of the number of samples collected.

| Method | | Unseen Platform | | |
| --- | --- | --- | --- | --- |
| | | Raspi4 | ASIC | FPGA |
| MAML [12] | | 0.568 | 0.602 | 0.541 |
| ANP [14] | | 0.801 | 0.657 | 0.884 |
| Meta-SGD [19] | | 0.844 | 0.831 | 0.882 |
| **HELP (Ours)** | Amortization | 0.568 | 0.604 | 0.539 |
| | + HW-Condition | 0.853 | 0.904 | 0.861 |
| | + Few-Shot Adapt | 0.872 | 0.913 | 0.866 |
| | + $z$ Modulator | **0.885** | **0.942** | **0.889** |

Table 2: The correlation of the estimated latency with HELP to the actual latency, on unseen devices with 10 measurement samples from each device (FBNet space).

**Computational Complexity of HELP.** The meta-training of the latency predictor is done only once, and once done, we can adapt the meta-latency estimator for the latency estimation of **any number of devices**. Since conventional approaches require to collect a large number of samples and train a device-specific latency estima-

| Method | Architecture Search Cost | Latency Estimator Building Cost |
| --- | --- | --- |
| Task-specific NAS | O(D) | O(DN) |
| MetaD2A [16] | O(1) | O(DN) |
| **MetaD2A + HELP** | **O(1)** | **O(D)** |

Table 1: Cost of NAS and latency estimation.

tor for each target device, while HELP only needs to collect 10 samples per device. HELP reduces the time complexity of obtaining latency estimations from $O(DN)$ to $O(D)$, where D is the number of devices and N is the number of samples to sufficiently train each latency estimator.

## 4 Experiment

In this section, we first verify the efficacy of our meta-learning scheme on the latency prediction of architectures on unseen devices, in Section 4.1. In this section, we also validate the sample-efficiency and the performance of HELP against existing latency prediction models, and a predictor trained with conventional meta-learning. Then, in Section 4.2, we validate HELP's effectiveness and efficiency on the full latency-constrained NAS for novel devices, by combining it with existing NAS methods.

**Search Space** Following the evaluation procedure of HW-NAS-Bench [18], we consider two search spaces, **NAS-Bench-201** [9] and **FBNet** [34]. Additionally, we consider **MobileNetV3** [13, 4] and **HAT** [33] search space for end-to-end latency-constrained NAS in Section 4.2. For a detailed description, refer to the supplementary file.

**Meta-Training Pool/Meta-Test Pool** To construct the **Meta-Training Pool**, we collect the latency measurements from **18 heterogeneous devices**, including GPUs, CPUs, mobile devices (NVIDIA 1080ti, Titan X, Titan XP, RTX 2080ti, Xeon Silver 4114, Silver 4210r, Samsung A50, S7, Google Pixel3, Essential Ph 1). For GPUs, we consider three different batch sizes [1, 32, 256(64)] and for all other hardware devices, we use the batch size of 1. We collect 900/4000 (architecture, latency) pairs of each training device for NAS-Bench-201 and FBNet search space, respectively. As for the **Meta-Test Pool**, we consider **Unseen Devices** and **Unseen Platforms**. Unseen Devices include NVIDIA GPU Titan RTX, Intel CPU Xeon Gold 6226, and Google Pixel2, which are different from the devices in the meta-training pool but belong to the same categories (GPU, CPU, mobile device). On the other hand, Unseen Platforms include Jetson AGX Xavier, Raspi4, ASIC-Eyeriss, and FPGA, which are completely unseen categories of devices. For Raspi4, ASIC-Eyeriss, FPGA and Pixel3, we use the latency measurements provided in the HW-NAS-Bench [18]. For the implementation details of our model, please refer to the supplementary file.

**Baselines** We compare our framework against relevant baselines. 1) **MAML** [12]: A few-shot regression baseline which meta-learns the initial parameters over multiple tasks via bi-level optimization. 2) **Meta-SGD** [19]: An extension of MAML with the meta-learned learning rate for the inner-gradient step. 3) **ANP** [14]: A few-shot regression model implemented with Attentional Neural Processes, which uses differentiable attentions to attend to the relevant contexts for the given query. 4) **BRP-NAS**[†] [10]: A predictor-based NAS method with a graph convolution neural network-based latency predictor. This baseline achieves the previous state-of-the-art performance on latency prediction in the NAS-Bench-201 search space. 5) **MetaD2A** [16]: This is a meta-NAS framework without a latency predictor which enables rapid architecture search (a few GPU seconds) for unseen datasets, which obtains state-of-the-art performance on multiple datasets in the NAS-Bench-201 search space.

---

[†]For experiment on LatBench provided by BRP-NAS, please refer to the supplementary file.

Table 3: Comparison of the latency estimators on unseen devices and platforms on NAS-Bench-201.

| Method | Transfer | Sample | Unseen Device | | | Unseen Platform | | | Mean |
|---|---|---|---|---|---|---|---|---|---|
| | | | GPU | CPU | Pixel2 | Raspi4 | ASIC | FPGA | |
| FLOPS | | - | 0.950 | 0.826 | 0.765 | 0.846 | 0.437 | 0.900 | 0.787 |
| Layer-wise Predictor | | - | 0.667 | 0.866 | - | - | - | - | 0.767 |
| BRP-NAS [10] | | 900 | 0.814 | 0.796 | 0.666 | 0.847 | 0.811 | 0.801 | 0.789 |
| BRP-NAS(+extra samples) | | 3200 | 0.822 | 0.805 | 0.693 | 0.853 | 0.830 | 0.828 | 0.805 |
| **HELP (Ours)** | ✓ | **10** | **0.987** | **0.989** | **0.802** | **0.890** | **0.940** | **0.985** | **0.932** |

(a) Titan RTX GPU          (b) Intel Xeon Gold CPU

Figure 4: Comparison of the **estimated** and **measured** latencies on a Titan RTX GPU and Intel Xeon Gold CPU. While BRP-NAS requires 900 samples to train the latency predictor, our meta-latency predictor requires only 10 samples for adaptation, and significantly outperforms it in the estimation performance.

## 4.1 Efficacy of HELP on Few-shot Latency Estimation for Novel Devices

We first validate whether the transferring the meta-knowledge obtained over a meta-training pool to an unseen meta-test device helps to improve the sample efficiency and prediction performance of the latency predictor. We adapt the meta-latency predictor on 10 architecture-latency pairs of 6 unseen meta-test devices, and report the Spearman's rank correlation (higher the better) between the estimated latencies and actual latencies on 1,000 neural architectures from the test set in FBNet search space, over 5 random runs. (Figure 3, Table 2, and Figure 5).

**Latency Estimation Performance for Unseen Devices** Figure 3 reports the average value of correlation scores of different predictors as a function of the number of architecture-latency pairs from meta-test devices. Shaded regions are the range of standard deviation of 5 runs with random seeds. HELP and Meta-SGD uses the initial parameters that are meta-learned over a large meta-training pool, and fine-tunes its parameter with given training samples. On the other hand, Scratch means to simply train a regression model from scratch with a given samples. The result shows that using meta-knowledge (HELP and Init. of Meta-SGD) consistently outperforms the model trained from scratch. Specifically, HELP's latency predictor with hardware-adaptive initial parameter $\theta_0$ achieves significantly larger performance gain over baselines when the number of samples is smaller (e.g., 10 and 50). Such sample-efficiency allows HELP to search for architectures that satisfy the latency constraints with orders of magnitudes shorter time compared to existing methods.

**Effect of the Hardware-adaptive Meta-learning** We analyze the effect of hardware-adaptive meta-learning in Table 2. We observe that the meta-latency predictor using the proposed modules largely outperforms the other ones trained with other meta-learning baselines that are hardware-independent, which shows the effectiveness of our hardware-adaptive meta-learning on unseen platforms. This is due to the heterogeneity of the tasks (devices) in the meta-training dataset, in which the task-conditioning becomes more important.

**Effect of the Size of the Device Pool** We further analyze the effect of the size of the meta-training pool, on the performance of the meta-latency predictor. Figure 5 reports the performance of our latency predictor over different sizes of the randomly sampled device pool. In particular, when the number of devices in the meta-training pool is 10 or more, our model achieves over 0.9 correlation on unseen devices using only 10 samples of the unseen target device, regardless of the device types of the meta-training pool. Contrarily, Meta-SGD does not yield meaningful performance gains even with large meta-training pools.

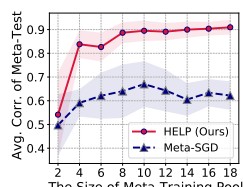

Figure 5: Effect of the meta-training pool size.

**Sample-efficiency of HELP** To demonstrate the sample-efficiency of our meta-learned latency predictor, we compare against three baselines: 1) a proxy predictor using number of FLOPs 2) a layer-wise predictor and 3) the latency predictor from BRP-NAS [10] (Table 3 and Figure 4). The results show that FLOPs, although easy to compute, is an inaccurate proxy for latency estimation.

Table 4: Performance comparison of different latency estimators combined with MetaD2A for latency-constrained NAS, on CIFAR-100 dataset with NAS-Bench-201 search space. For the building time and the total NAS cost of MetaD2A+HELP, we report only time and cost during the meta-test time. The meta-training time of HELP is 25 hours and the time to meta-train the MetaD2A is 46 GPU hours, which is conducted only once across all unseen devices.

| Device | Model | Const (ms) | Latency (ms) | Accuracy (%) | MACs (M) | Sample | Building Time | Total NAS Cost (Wall Clock) | Speed Up |
|---|---|---|---|---|---|---|---|---|---|
| Unseen Device Google Pixel2 | MetaD2A + BRP-NAS [10] | 14 | 14 | 66.9 | 79 | 900 | 1120s | 1220s | 1.0× |
| | MetaD2A + HELP (Ours) | | 13 | 67.4 | 47 | **10** | **25s** | **125s** | **9.8×** |
| | MetaD2A + BRP-NAS [10] | 22 | 34 | 73.5 | 185 | 900 | 1120s | 1220s | 1.0× |
| | MetaD2A + HELP (Ours) | | 19 | 70.6 | 55 | **10** | **25s** | **125s** | **9.8×** |
| | MetaD2A + BRP-NAS [10] | 34 | 34 | 73.5 | 185 | 900 | 1120s | 1220s | 1.0× |
| | MetaD2A + HELP (Ours) | | 34 | 73.5 | 185 | **10** | **25s** | **125s** | **9.8×** |
| Unseen Device Titan RTX (Batch 256) | MetaD2A + Layer-wise Pred. | 18 | 37 | 73.2 | 121 | | 998s | 1098s | 1.0× |
| | MetaD2A + BRP-NAS [10] | | 21 | 67.0 | 86 | 900 | 940s | 1040s | 1.1× |
| | MetaD2A + HELP (Ours) | | 18 | 69.3 | 51 | **10** | **11s** | **111s** | **9.9 ×** |
| | MetaD2A + Layer-wise Pred. | 21 | 41 | 73.5 | 184 | | 998s | 1098s | 1.0× |
| | MetaD2A + BRP-NAS [10] | | 19 | 71.5 | 55 | 900 | 940s | 1040s | 1.1× |
| | MetaD2A + HELP (Ours) | | 19 | 71.6 | 55 | **10** | **11s** | **111s** | **9.9×** |
| | MetaD2A + Layer-wise Pred. | 25 | 41 | 73.5 | 184 | | 998s | 1098s | 1.0× |
| | MetaD2A + BRP-NAS [10] | | 23 | 70.7 | 82 | 900 | 940s | 1040s | 1.1× |
| | MetaD2A + HELP (Ours) | | 25 | 71.8 | 86 | **10** | **11s** | **111s** | **9.9×** |

Figure 6: The accuracy-latency trade-off of the NAS framework with an oracle accuracy predictor, combined with the oracle latency estimator (yellow star), BRP-NAS (green square), layer-wise predictor (blue triangle), and HELP (Ours - red circle), on various devices in NAS-Bench-201 space. HELP, with its accurate latency estimation, obtains Pareto-frontier models, while baselines yield sub-optimal architectures.

The layer-wise predictor also achieves poor performance, because it cannot reflect the complexity and the holistic effect of the network architecture. The latency predictor from BRP-NAS [10], which is a 4-layer GCN with 600 hidden units followed by a fully connected layer to produce a scalar output, achieves significantly better estimation compared to the first two baselines. However, this model requires 900 samples from each architecture-device pair, as described in [10]. Finally, our HELP predictor achieves the best performance, achieving the Spearman's rank correlations of 0.987 on GPU and 0.989 on CPU, using only 10 latency measurements of the architecture on each device. This shows the clear advantage of our method, in terms of the estimation accuracy and sample-efficiency.

## 4.2 End-to-end Latency-constrained NAS with HELP

To show that HELP does help NAS frameworks rapidly obtain latency-constrained/optimal architectures for a novel device, we combine HELP with existing NAS methods, namely MetaD2A [16], OFA [4] and HAT [33], and validate the performance on the latency-constrained NAS tasks.

In Table 4 and Table 5, besides latency and accuracy, we additionally report the time-efficiency of the latency estimators with three different measures. First, we use the **number of samples** which are the number of architecture-latency pairs obtained from the *target* device, that are used to build or train the latency estimator. We also report the **building time**, which is the total wall clock time to build the latency estimator, including the time required for sample collection, architecture compilation on the target devices, transmitting the architecture to the target device, and measuring the latency on the device. Finally, we report **the total NAS cost**, which is the sum of both the estimator building time and the architecture search time, on a target task. After obtaining the architecture, we measure the actual latency of the architecture on the target device and report it as **latency** (ms). For the building time and the total NAS cost, we exclude the cost of any procedures that are not done during the meta-test time, such as the meta-training of MetaD2A model (46 GPU hours) and HELP (25/18 hours for MetaD2A and OFA), as well as the time to train the supernet for OFA (1,200 GPU hours).

We first combine HELP with MetaD2A and compare it with the MetaD2A combined with other latency predictors. We conduct NAS on NAS-Bench-201 [9] benchmark for the CIFAR-100 dataset,

Table 5: The results of the latency constrained-NAS experiment for ImageNet-1k with MobileNetV3 search space. For the building time and the total NAS cost of OFA+HELP, we report only time and cost during the meta-test time. The meta-training time of HELP is 18 hours and the time to train the supernet for OFA is 1,200 GPU hours, which is conducted only once across all unseen devices.

| Device | Model | MACs (M) | Latency (ms) | Accuracy (%) | Latency Model Sample | Latency Model Building Time | Total NAS Cost (Wall Clock) | Speed Up |
|---|---|---|---|---|---|---|---|---|
| Unseen Device Titan RTX (Batch 64) | MobileNetV3-Large [13] | 219M | 22.1 | 75.2 | - | - | - | - |
| | MnasNet-A1 [29] | 312M | 20.0 | 75.2 | 8k | 4.5h | 40,004.5h | 1.0× |
| | FBNet-C [34] | 375M | 27.5 | 74.9 | 7.5k | 4.2h | 580.2h | 69× |
| | ProxylessNAS-GPU [3] | 465M | 22.0 | 75.1 | 5k | 2.8h | 502.8h | 80× |
| | OFA+Layer-wise Pred. [4] | 397M | 21.5 | 76.4 | 27k | 15h | 15h | 2667× |
| | OFA + HELP (20ms) | 230M | 20.3 | 76.0 | | | | |
| | OFA + HELP (23ms) | 268M | 23.1 | 76.8 | **10** | **26s** | **0.007h (26s)** | **5.7M×** |
| | OFA + HELP (28ms) | 346M | 28.6 | 77.9 | | | | |
| Unseen Platform Intel Xeon Gold 6226 | MobileNetV3-Large [13] | 219M | 132 | 75.2 | - | - | - | - |
| | MnasNet-A1 [29] | 312M | 212 | 75.2 | 8k | 35.6h | 40,035.6h | 1.0× |
| | FBNet-B [34] | 295M | 212 | 74.1 | 7.5k | 33.3h | 609.3h | 66× |
| | ProxylessNAS-CPU [3] | 465M | 200 | 75.1 | 5k | 22.2h | 522.2h | 77× |
| | OFA+Layer-wise Pred. [4] | 301M | 167 | 74.6 | 27k | 120h | 120h | 334× |
| | OFA + HELP (170ms) | 336M | 147 | 77.6 | **20** | **300s** | **0.08h (300s)** | **0.5M×** |
| | OFA + HELP (190ms) | 375M | 171 | 78.1 | | | | |
| Unseen Platform Jetson AGX Xavier (Batch 16) | MobileNetV3-Large [13] | 219M | 70.8 | 75.2 | - | - | - | - |
| | MnasNet-A1 [29] | 312M | 71.6 | 75.2 | 8k | 24.9h | 40,024.9h | 1.0× |
| | ProxylessNAS-GPU [3] | 465M | 82.6 | 75.1 | 5k | 15.6h | 515.6h | 78× |
| | OFA+Layer-wise Pred. [4] | 349M | 69.2 | 75.8 | 27k | 84h | 84h | 476× |
| | OFA + HELP (65ms) | 243M | 67.4 | 75.9 | **10** | **112s** | **0.03h (112s)** | **1.3M×** |
| | OFA + HELP (70ms) | 279M | 76.4 | 76.7 | | | | |

using Google Pixel2 mobile phone and NVIDIA Titan RTX GPU as the target devices[‡]. The results in Table 4 show that HELP largely outperforms BRP-NAS [10], the previous state-of-the art latency predictor, with 90× sample efficiency and 9.8×, 9.9× computational efficiency on Pixel2 and Titan RTX, respectively. Specifically, HELP + MetaD2A can efficiently retrieve an optimal latency-constrained architecture in 125s/111s for the given dataset on Pixel2/Titan RTX, respectively, while BRP-NAS [10]'s predictor with large building time (1120s/940s) becomes a bottleneck for MetaD2A's rapid NAS process. Further, we validate the accuracy-efficiency trade-off of HELP against the baseline latency estimators, by combining them with the oracle accuracy predictor on NAS-Bench-201 space (in Figure 6). With HELP, oracle NAS obtains near Pareto-optimal models in most cases, while combining it with other latency estimators yield sub-optimal models.

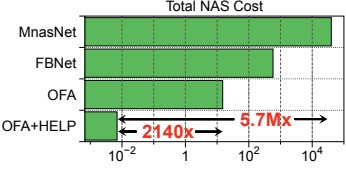

Figure 7: HELP reduces the total NAS cost by 2140× on Titan RTX. The total NAS cost is represented on a log-scale.

In Table 5, we validate HELP on a large-scale dataset, ImageNet [7], against state-of-the-art NAS methods. In this experiment, we only consider GPUs with various batch sizes (1, 32, 64) as the meta-training devices, and Titan RTX GPU, Xeon CPU and Jetson Edge GPU as unseen devices. We combine HELP with OFA by training the accuracy predictor on the pre-trained OFA network, and conducting predictor-guided evolutionary architecture search to fit the latency constraint. The result shows that, while searching for competitive architectures, HELP significantly reduces the total NAS cost of the baseline NAS methods which builds a new latency estimator for each target device. Figure 7 shows that OFA+HELP reduces the total NAS cost on a target device by 2140×, when compared with the OFA + layer-wise predictor. This allows us to benefit from the rapid search speed of OFA, since the total NAS cost is only tens or hundreds of seconds, while the OFA + layer-wise predictor takes 15 hours, which is impractical. The details of the searched architectures are provided in the supplementary file.

ProxylessNAS has roughly 42 unique blocks (7 different operations per 6 different input shapes) and use 5k architecture samples to build a layer-wise latency predictor. Thus, we proportionally estimate the number of samples to train its latency estimator for the FBNet and OFA supernet as 7.5k and 27k, which have 63 and 225 unique blocks, respectively. MnasNet does not have a latency estimator, but directly measures the latency of every architecture in the search process (8k architectures).

**Hardware-aware Transformer Architecture Search** To demonstrate the task-level generality of HELP, we further conduct end-to-end latency-constrained Transformer architecture search experi-

---
[‡]For more results on various devices, please check the supplementary file.

ments on machine translation task, WMT'14 En-De, by combining HELP with HAT [33], which is hardware-aware NAS method for Transformer. For WMT'14 En-De, we follow [33, 35] for training,

Table 6: Results of hardware-aware Transformer architecture search on WMT'14 En-De. By combining HAT with HELP, 200× fewer samples are used to train latency predictor while achieving competitive performance.

| Target Device | Model | Constraint | Latency | Number of Samples | BLEU score |
|---|---|---|---|---|---|
| Unseen Device GPU NVIDIA Titan RTX | HAT+End-to-End Pred. | 90ms | 73.9ms | 2000 | 27.08 |
| | HAT+HELP (Ours) | 90ms | 74.0ms | **10** | 27.19 |
| | HAT+End-to-End Pred. | 150ms | 108.4ms | 2000 | 27.04 |
| | HAT+HELP (Ours) | 150ms | 106.5ms | **10** | 27.44 |
| Unseen Platform CPU Intel Xeon Gold6240 | HAT+End-to-End Pred. | 200ms | 159.6ms | 2000 | 27.20 |
| | HAT+HELP (Ours) | 200ms | 159.6ms | **10** | 27.20 |
| | HAT+End-to-End Pred. | 400ms | 369.4ms | 2000 | 28.09 |
| | HAT+HELP (Ours) | 400ms | 343.2ms | **10** | 27.52 |

validation, test setting of datasets. The meta-training device pool is configured only with GPUs such as NVIDIA Titan X, 1080ti, 2080ti, and the unseen devices are Titan RTX GPU and Intel Xeon CPU. As a baseline model, we train the end-to-end latency predictor (End-to-End Pred.) using 2000 architecture-latency pair samples for each device following HAT [33]. Table 6 shows the results of NAS with different latency constraints, and BLEU score of searched models. HAT+HELP, which replaces a latency predictor of HAT with HELP, successfully obtains the competitive Transformer models while using 200× fewer samples for training latency predictor than the original HAT.

## 5 Discussion

**Limitation** The proposed hardware-conditioned meta-learning framework allows HELP obtain an optimal latency-constrained network within few seconds, when combined with rapid NAS methods such as [16, 4, 33], since building a latency estimator is often a bottleneck for them. However, combining HELP with slower NAS methods based on RL, gradient-based search, and evolutionary algorithms will be less effective since the total NAS cost is dominated by the architecture search cost, rather than the time required to build the latency estimator. Yet, since mainstream NAS research nowadays is focusing more on the reduction of the architecture search cost [25, 22, 8, 4, 16], we believe that the latency estimation will become more of a bottleneck, and our sample-efficient latency estimator will become even more useful for latency-constrained NAS.

**Societal Impact** Since our method requires to build a meta-training pool only once, and meta-latency estimator can rapidly adapt to a new device with as few as 10 samples, we can largely reduce the waste in the computational resources required for obtaining the latency measurements. Since the repeated measurements require large energy consumption that also yields high $CO_2$ emissions, and reduce devices' lifetime, our method is more environment-friendly than existing methods that require a large number of measurements from each device.

## 6 Conclusion

We proposed a novel meta-learned latency predictor, that can estimate the latency of an architecture on a novel device, using only a few measurements from it. While conventional latency prediction methods are inefficient since they cannot generalize across devices, and require a large number of latency measurements for each device, our latency predictor is meta-learned to rapidly adapt to an unseen device by utilizing the meta-knowledge accumulated over a device pool. Using a novel hardware embedding function that embeds each device based on its latencies on a set of reference architectures, we conducted hardware-conditioned meta-learning to obtain a device-specific initial parameters, and further took inner gradient steps to adapt to a new device. We validated our meta-latency predictor by measuring its latency estimation performance on unseen devices, on which it outperforms baselines, using only 10 to 20 samples per device. Furthermore, we combined our latency predictor with three rapid NAS methods, to show that it performs latency-constrained NAS on unseen devices extremely fast and accurately.

**Acknowledgements** This work was supported by Institute of Information & communications Technology Planning & Evaluation (IITP) grant funded by the Korea government (MSIT) (No.2019-0-00075), Samsung Research Funding Center of Samsung Electronics (No. IO201214-08145-01, IO201210-08006-01), and the Engineering Research Center Program through the National Research Foundation of Korea (NRF) funded by the Korean Government MSIT (NRF-2018R1A5A1059921). We thank Seul Lee for providing helpful feedbacks in

preparing an earlier version of the manuscript and NMSL laboratory of KAIST for supporting various mobile devices. We also thank the anonymous reviewers for their insightful comments and suggestions.

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
