**Organization**   In this supplementary file, we provide in-depth descriptions of the materials that are not covered in the main paper, and report additional experimental results. The document is organized as follows:

- **Section A** - We elaborate on the detailed *experiment setups*, such as search space, reference devices, reference architectures, and latency measurement pipeline.
- **Section B** - We provide the *implementation and training details*, such as model structure, learning rate and hyper-parameters of learning the proposed hardware-adaptive latency prediction model.
- **Section C** - We provide the results of *additional experiments* and visualization of the obtained architectures on different devices.

# A   Experimental Setups

## A.1   Search Space

**NAS-Bench-201** search space contains cell-based neural architectures which represent an architecture as a graph, where each cell consists of 4 nodes and 6 edges and each edge has 5 operation candidates, such as zerorize, skip connection, 1-by-1 convolution, 3-by-3 convolution, and 3-by-3 average pooling, which leads to the total of 15626 unique architectures. The macro skeleton is stacked with one stem cell, three stages of 5 repeated cells each, residual blocks [35] between the stages, and final classification layer consisting of a average pooling layer and a fully connected layer with softmax function. The stem cell consists of a 3-by-3 convolution with 16 output channels followed by a batch normalization layer [38], each cell with three stages has 16, 32 and 64 output channels, respectively. The intermediate residual blocks have convolution layers with the stride 2 for down-sampling.

Table A.1: Configurations of 9 candidate blocks of FBNet search space.

| Candidates | Kernel size | Expansion ratio | Group |
|---|---|---|---|
| k3_e1 | 1 | 3 | 1 |
| k3_e1_g2 | 1 | 3 | 2 |
| k3_e3 | 3 | 3 | 1 |
| k3_e6 | 6 | 3 | 1 |
| k5_e1 | 1 | 5 | 1 |
| k5_e1_g2 | 1 | 5 | 2 |
| k5_e3 | 3 | 5 | 1 |
| k5_e6 | 6 | 5 | 1 |
| skip | - | - | - |

**FBNet** search space is a layer-wise space with a fixed macro-architecture, which is choosing a building block among 9 pre-defined candidates per 22 unique positions and the rest is fixed, resulting in $9^{22} \approx 10^{21}$ unique architectures. The block structure is inspired by MobileNetV2 [53] and ShiftNet [62], which adopts both mobile-block convolutions (MBConv) and group convolution. The configurations of 9 candidate blocks are provided in Table A.1.

**MobileNetV3** search space is also layer-wise space, where a building block adopts MBConvs, squeeze and excitation [37], and modified swish nonlinearity to build more efficient neural network. The search space consists of 5 stages, and in each stage, the number of building blocks ranges across $\{2, 3, 4\}$. For each block, the kernel size should be chosen from $\{3, 5, 7\}$, and the expansion ratio should be chosen from $\{3, 4, 6\}$. This leads the search process to a choice out of around $10^{19}$.

## A.2   Reference Device Details and Latency Measurement Pipeline

Table A.2: Specifications of reference devices used in the paper. We consider 17 hardware devices from 7 representative platforms such as Desktop GPU, Edge GPU, Server CPU, Mobile phone, Raspberry Pi, ASIC, and FPGA.

| Platform | Name | Micro Architecture | The Number of Cores | Memory |
|---|---|---|---|---|
| Desktop GPU | NVIDIA GTX 1080ti [8] | Pascal | CUDA 3584 | 11GB GDDR5 |
| | NVIDIA GTX Titan X [11] | Pascal | CUDA 3584 | 12GB GDDR5 |
| | NVIDIA GTX Titan XP [12] | Pascal | CUDA 3840 | 12GB GDDR5 |
| | NVIDIA RTX 2080ti [9] | Turing | CUDA 4352 | 11GB GDDR6 |
| | NVIDIA Titan RTX [10] | Turing | CUDA 4608 | 24GB GDDR6 |
| Edge GPU | NVIDIA Jetson AGX Xavier [6] | Volta | CUDA 512 | 32GB LPDDR4 |
| Sever CPU | Intel Xeon Silver 4114 [4] | Intel P6 | CPU Core 10 | Cache 13.75MB |
| | Intel Xeon Silver 4210r [5] | Intel P6 | CPU Core 10 | Cache 13.75MB |
| | Intel Xeon Gold 6226 [3] | Intel P6 | CPU Core 12 | Cache 19.25MB |

| Platform | Name | SoC | The Number of CPU Cores | Memory |
|---|---|---|---|---|
| Mobile Phone | Samsung Galaxy A50 [16] | Samsung Exynos 9610 | 4 Cortex-A73 & 4 Cortex-A53 | 4GB LPDDR4 |
| | Samsung Galaxy S7 [17] | Samsung Exynos 8890 | 4 Exynos M1 & 4 Cortex-A53 | 4GB LPDDR4 |
| | Essential PH-1 [13] | Qualcomm Snapdragon 835 | 8 Kryo 280 | 4GB RAM |
| | Google Pixel2 XL [14] | Qualcomm Snapdragon 835 | 8 Kryo 280 | 4GB LPDDR4X |
| | Google Pixel3 [15] | Qualcomm Snapdragon 845 | 8 Kryo 385 | 4GB LPDDR4X |
| Raspberry Pi | Raspberry Pi 4 [43] | Broadcom BCM2711 | 4 ARM A72 | 4GB LPDDR4 |
| ASIC | Eyeriss [43] | Please refer to descriptions of Section A.2 | | |
| FPGA | FPGA [43] | Please refer to descriptions of Section A.2 | | |

In this paper, we consider a wide variety of hardware devices of 7 representative platforms such as Desktop GPU, Edge GPU, Server CPU, Mobile Phone, Raspberry Pi, ASIC, and FPGA. Each device has different hardware structure and specification even in the same hardware platform, thus, we report hardware specification of all hardware devices except ASIC-Eyeriss and FPGA in Table A.2. In the case of ASIC, we use Eyeriss, which is a state-of-the-art accelerator [26] for deep CNNs. For FPGA, we use Xilinx ZC706 board with the Zynq XC7Z045 SoC which includes 1 GB DDR3 memory SODIMM [7]. For 4 devices such as Google Pixel3, Raspberry Pi 4, ASIC-Eyeriss, and FPGA, we use the latency data provided from HW-NAS-Bench [43], and for other devices, we directly measure the latency values to collect meta-training data on three search space such as NAS-Bench-201 space, FBNet space, and MobileNetV3 space.

In addition, we briefly describe how the latency data were collected in HW-NAS-Bench, please refer to original paper [43] for the details. In the case of ASIC-Eyeriss, the latency values are estimated from two simulator, Accelergy [65]+Timeloop [50] and DNN-Chip Predictor [71], each automatically identifying the optimal algorithm-to-hardware mapping methods for the architecture. For Raspi 4, all architectures are converted into Tensorflow Lite [20] (TFLite) format and executed with the official interpreter which is preconfigured in the Raspi 4. Similarly, all architectures are converted into TFLite format, and the official benchmark binary files are used for latency measurement on Pixel 3. For the last, to obtain the latency on FPGA, they implement a chunk-based pipeline structure [55, 70] and compile architectures using the Vivado HLS toolflow [1].

Now, we describe the latency measurement pipeline for desktop GPUs, Jetson, server CPUs, and mobile phone. Note that, throughout all hardware devices, latency data is collected averaging 50 times after 50 times initial runs to activate the device.

**Desktop GPU**, **Jetson** and **Server CPU**: To directly measure the latency of architectures from considered search space and baselines on these devices, the neural networks are implemented with PyTorch 1.8.1 [51] and executed with cuDNN [27] for desktop GPUs and Jetson, and with MKL-DNN for server CPUs.

**Mobile Phone**: (1) We load a neural architecture from architecture configuration with PyTorch [51] framework. (2) We serialize the neural architecture using `TorchScript` library [19]. (3) We build Android application [2] with `PyTorch Android` library 1.9 [18] that measures the inference time of a serialized architecture on a target mobile phone and collects latency data by running the application.

**Measurement Details** We used latencies provided from the HW-NAS-Bench dataset itself for FGPA, ASIC, Raspi4, Pixel3. For other devices, we directly measured latencies, by discarding the first ten

measurements and removing the top 10% and the bottom 10% values. Then we measured the latency of architecture on the target device 50 times and averaged the results.

To collect layer-wise latency data, following [23], we sample appropriate numbers of full architectures and run each full architecture while recording the time taken for each block type and input image size, not run each block piece respectively. A latency of each block and input image size pair is averaged over the number of sampled architectures which contains it, as well as 50 time runs.

## A.3 Correlation between Devices

Table A.3: Spearman's rank correlation coefficient of collected latencies among 8 representative devices in NAS-Bench-201 search space. 1080ti_1 means 1080ti GPU with batch size 1.

| Device | 1080ti_1 | 1080ti_256 | 2080ti_1 | 2080ti_256 | Silver4114 | Gold6226 | Pixel2 | Pixel3 |
|---|---|---|---|---|---|---|---|---|
| 1080ti_1 | 1.00 | 0.74 | 0.96 | 0.78 | 0.83 | 0.96 | 0.80 | 0.59 |
| 1080ti_256 | | 1.00 | 0.72 | 0.82 | 0.76 | 0.78 | 0.77 | 0.74 |
| 2080ti_1 | | | 0.99 | 0.76 | 0.81 | 0.94 | 0.79 | 0.57 |
| 2080ti_256 | | | | 1.00 | 0.95 | 0.88 | 0.88 | 0.87 |
| Silver4114 | | | | | 0.98 | 0.91 | 0.87 | 0.81 |
| Gold6226 | | | | | | 0.97 | 0.86 | 0.71 |
| Pixel2 | | | | | | | 0.94 | 0.76 |
| Pixel3 | | | | | | | | 1.00 |

To show how much the latencies of the same architectures on different devices are correlated, we report the Spearman's rank correlation coefficient among measured ground truth latencies of multiple hardware devices from each hardware platform on the same set of architectures in Table A.3. To reflect a measurement error, we collect two sets of latencies to compute the correlation coefficient of the device itself. As shown in Table A.3, the measured latencies on the same set of architectures are largely different (e.g., pixel2 vs pixel3, GPU 1080ti vs 2080ti with batch size 256). Furthermore, even with the same GPU device, the correlation scores are not high if the batch sizes are different. Therefore, the result shows that one cannot simply use a latency estimator from a device for a latency estimation of another device from the same platform and expect it to have high accuracy.

## A.4 Reference Architecture Details

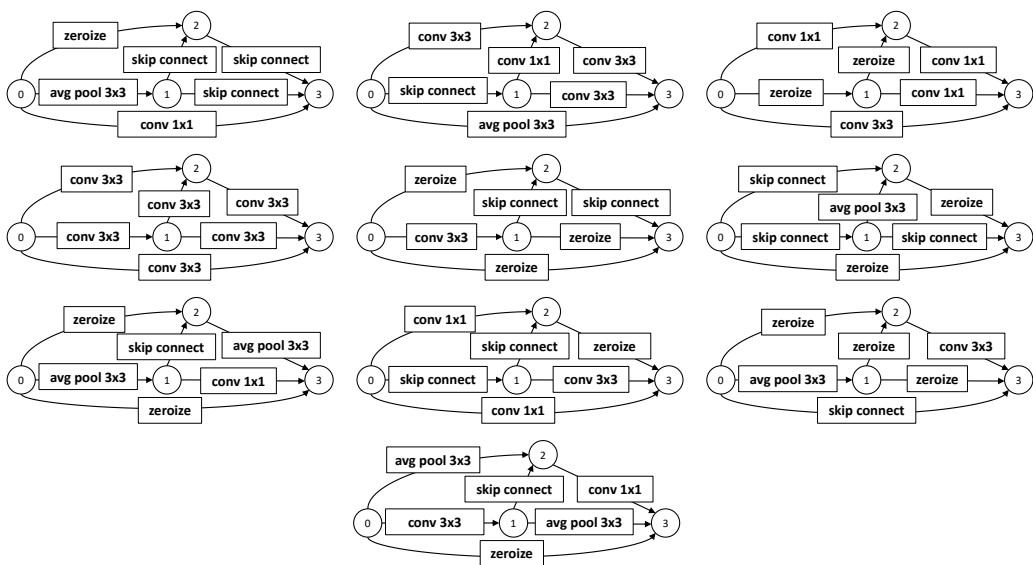

Figure A.1: Visualization of 10 reference neural architectures we used for NAS-Bench-201 search space. Architecture indices of NAS-Bench-201 are 11982, 13479, 14451, 1462, 431, 55, 6196, 8636, 9, 9881 in order of left top to right bottom.

To handle various devices with a single prediction model, we introduced reference neural architectures that enable hardware condition prediction, in Section 3.2 of the main paper. We considered a

device as a black-box function that takes such reference architectures as an input and outputs a latency set of reference architectures as a hardware embedding. We randomly selected 10 reference architectures for each search space (NAS-Bench-201, FBNet, and MobileNetV3) and used them across all experiments and devices of the same search space. In Figure A.1, we visualize 10 reference architectures that we used in NAS-Bench-201 search space. Reference architectures have diverse structures as shown in Figure A.1 and their latency values cover a wide range, for example, latency values of such reference architectures measured on NVIDIA Titan RTX (batch 256) are $\{5.9, 8.5, 11.8, 14.1, 15.5, 16.6, 17.6, 19.1, 28.5, 44.3\}$. Reference architecture indices in NAS-Bench-201 are 11982, 13479, 14451, 1462, 431, 55, 6196, 8636, 9, 9881. We include example reference architectures of all search spaces that we used in the experiments and their measured latency values on target devices, in the code and dataset that we submit.

# B  Implementation and Training Details

| Common Setting | |
| --- | --- |
| Meta-batch size | 8 |
| The number of inner update | 2 |
| The number of episode | 2000 |
| Dimension of hardware input | 10 |
| Dimension of hidden layer of architecture encoder | 100 |
| Dimension of hidden layer of device. encoder | 100 |
| Dimension of hidden layer of header | 200 |
| **NAS-Bench-201 search space** | |
| Meta-learning rate | 1e-4 |
| Dimension of architecture input | 8 |
| The number of GCN | 4 |
| **FBNet search space** | |
| Meta-learning rate | 1e-3 |
| Dimension of architecture input | 132 |
| **MobileNetV3 search space** | |
| Meta-learning rate | 1e-3 |
| Dimension of architecture input | 145 |

Table B.1: Hyperparameter settings of HELP

In this section, we describe the details of HELP implementation and hyperparameters that we used in the experiments. HELP consists of four main modules such as an architecture encoder, a device encoder, and a header for output, and an inference network for $z$. By combining the first three modules, we construct $f$ and the last module is $g$. While the architecture encoder has a different structure dependent on search space, all other modules have the same structure for all search spaces. Specifically, we use a 4-layers GCN to encode graph-based architectures of NAS-Bench-201 and use two multi-layer perceptrons (MLPs) to encode one-hot based flat topology of architectures of FBNet space and MobileNetV3 space. For the one-hot encoding, we follow OFA [24]. As the device encoder, we use two MLPs that take a latency set of reference architectures measured on a target device as an input and output a hardware embedding. By concatenating the architecture embedding and the hardware embedding, we feed it into the header that consists of three MLPs to output the estimated latency, in a scalar value. The inference network takes the reference latency set as an input and outputs scaling parameters with 603 dimensions which are equal to the number of weights and biases of the header. All dimension of hidden layers is 100 and we denote the hyperparameters for HELP as shown in Table B.1. We use Adam optimizer and mean square error as a loss function for all experiments.

# C   Additional Experiments

## C.1   Experiment on NAS-Bench-201 Search Space

Table C.1: Performance comparison of different latency estimators combined with MetaD2A for latency-constrained NAS, on CIFAR-100 dataset with NAS-Bench-201 search space. We use Eyeriss (top) and FPGA (middle) as unseen platforms and Xeon GPU (bottom) as a unseen devices. We exclude the layer-wise predictor for Eyeriss and FPGA since we use the latency measurements from HW-NAS-Bench [43] for this device, and it does not provide block information for the layer-wise predictor to use.

| Device | Model | Const (ms) | Latency (ms) | Accuracy (%) | Latency Model Sample | Efficiency |
|---|---|---|---|---|---|---|
| Unseen Platform ASIC-Eyeriss | MetaD2A + BRP-NAS [32] | 5 | $4.7_{\pm 0.8}$ | $71.7_{\pm 0.2}$ | 900 | $1.0\times$ |
| | MetaD2A + HELP (Ours) | | $4.1_{\pm 0.6}$ | $69.8_{\pm 1.9}$ | **10** | $\mathbf{90.0\times}$ |
| | MetaD2A + BRP-NAS [32] | 7 | $9.1_{\pm 0.0}$ | $73.5_{\pm 0.0}$ | 900 | $1.0\times$ |
| | MetaD2A + HELP (Ours) | | $5.5_{\pm 0.8}$ | $71.9_{\pm 0.2}$ | **10** | $\mathbf{90.0\times}$ |
| | MetaD2A + BRP-NAS [32] | 9 | $9.1_{\pm 0.0}$ | $73.5_{\pm 0.0}$ | 900 | $1.0\times$ |
| | MetaD2A + HELP (Ours) | | $9.1_{\pm 0.0}$ | $73.5_{\pm 0.0}$ | **10** | $\mathbf{90.0\times}$ |
| Unseen Platform FPGA | MetaD2A + BRP-NAS [32] | 5 | $7.2_{\pm 0.5}$ | $73.4_{\pm 0.2}$ | 900 | $1.0\times$ |
| | MetaD2A + HELP (Ours) | | $4.7_{\pm 0.0}$ | $71.8_{\pm 0.0}$ | **10** | $\mathbf{90.0\times}$ |
| | MetaD2A + BRP-NAS [32] | 6 | $7.4_{\pm 0.0}$ | $73.5_{\pm 0.0}$ | 900 | $1.0\times$ |
| | MetaD2A + HELP (Ours) | | $5.9_{\pm 0.0}$ | $72.4_{\pm 0.0}$ | **10** | $\mathbf{90.0\times}$ |
| | MetaD2A + BRP-NAS [32] | 7 | $7.4_{\pm 0.0}$ | $73.5_{\pm 0.0}$ | 900 | $1.0\times$ |
| | MetaD2A + HELP (Ours) | | $7.4_{\pm 0.0}$ | $73.5_{\pm 0.0}$ | **10** | $\mathbf{90.0\times}$ |
| Unseen Device Xeon CPU Gold 6226 | MetaD2A + Layer-wise Pred. | 8 | $6.2_{\pm 0.0}$ | $64.4_{\pm 0.0}$ | 900 | $1.0\times$ |
| | MetaD2A + BRP-NAS [32] | | $9.5_{\pm 0.5}$ | $66.9_{\pm 0.0}$ | 900 | $1.0\times$ |
| | MetaD2A + HELP (Ours) | | $7.7_{\pm 1.5}$ | $66.6_{\pm 0.0}$ | **10** | $\mathbf{90.0\times}$ |
| | MetaD2A + Layer-wise Pred. | 11 | $10.7_{\pm 0.0}$ | $70.2_{\pm 0.0}$ | 900 | $1.0\times$ |
| | MetaD2A + BRP-NAS [32] | | $8.7_{\pm 0.0}$ | $68.2_{\pm 0.0}$ | 900 | $1.0\times$ |
| | MetaD2A + HELP (Ours) | | $11.0_{\pm 0.6}$ | $70.6_{\pm 0.9}$ | **10** | $\mathbf{90.0\times}$ |
| | MetaD2A + Layer-wise Pred. | 14 | $14.1_{\pm 0.0}$ | $71.8_{\pm 0.0}$ | 900 | $1.0\times$ |
| | MetaD2A + BRP-NAS [32] | | $17.0_{\pm 0.0}$ | $73.5_{\pm 0.0}$ | 900 | $1.0\times$ |
| | MetaD2A + HELP (Ours) | | $13.9_{\pm 0.4}$ | $72.1_{\pm 0.0}$ | **10** | $\mathbf{90.0\times}$ |

In Table 4 of the main paper, we show how the proposed HELP effectively reduces NAS costs of latency-constrained NAS tasks on two representative devices such as Google Pixel2 phone and NVIDIA Titan RTX GPU with NAS-Bench-201 search space. By attaching meta-trained HELP to the rapid NAS method, MetaD2A, the total NAS costs on Pixel2 and Titan RTX are only 125s and 111s, respectively. As in Table 4, we validate the efficiency of HELP on additional various devices such as Eyeriss and FPGA as unseen platforms and Xeon CPU Gold 6226 as an unseen device in Table C.1. We exclude the layer-wise predictor for Eyeriss and FPGA since we use the latency measurements from HW-NAS-Bench [43] for this device, and it does not provide block information for the layer-wise predictor to use. We report an average of the results of 3 runs using random seeds for each experiment, with 95% confidence intervals. Following the experiment settings in the paper of BRP-NAS [32], we train the baseline models using 900 architecture-latency sample pairs per device. To meta-train our HELP model, we consider 7 devices as reference devices such as NVIDIA Titan 1080ti, Intel Xeon Silver 4114, Intel Xeon Silver 4210r, and Samsung A50 phone, Samsung S7 phone, Google Pixel3 phone, Essential Ph 1 phone and use batch size 1, 32, 256 for Titan 1080ti and 1 for other devices and collect 900 samples per device. After meta-training HELP with collected data samples, we rapidly adapt HELP to target devices (Eyeriss, FPGA, and Gold 6226) with 10 reference architecture-latency pairs measured on each target device.

As shown in Table C.1, HELP adapted only with 10 samples obtains neural architectures where latency measured on the target devices are closer to latency constraints (Const) than the baselines in 6 out of 9 cases across various devices. For example, for latency constraints 5 (ms) and 6 (ms) of FPGA, MetaD2A + HELP provides neural architectures with 4.7 (ms) and 5.9 (ms) latencies, respectively, while latencies of architectures obtained by MetaD2A + BRP-NAS exceed far from latency constraints, as 7.2 (ms) and 7.5 (ms), respectively. The proposed method searches for architectures that satisfy a given constraint by responding sensitively even if the interval between

constraints is small. Similarly with the results of FPGA, for the unseen device, Intel Xeon CPU Gold 6226 with constraints 8 (ms), 11 (ms), 14 (ms), MetaD2A + HELP provides architectures that meet constraints such as 7.7 (ms), 11.0 (ms), and 13.9 (ms), respectively, and have high performance. The obtained architectures by the baselines either exceed constraints or have lower performance than ours. For various devices, HELP shows reliable prediction with only 10 measurements on a target device in the latency-constrained NAS tasks.

Table C.2: Mean of Spearman's rank correlation coefficient when the meta-training devices and the meta-test devices of Table 3 in the main paper are swapped.

| Methods | FLOPs | Layer-wise Predictor | BRP-NAS [32] | BRP-NAS (+extra samples) | HELP (Ours) |
|---|---|---|---|---|---|
| Samples from Target Devices | - | - | 900 | 3200 | **10** |
| Mean Corr. Coeff. | 0.747 | 0.858 | 0.773 | 0.793 | **0.940** |

In Table 3 in the main paper, we report the correlation between real-measured latency and latency estimated by HELP that meta-trained with 18 devices for 6 target devices. In this experiment, we swap meta-training devices and meta-test devices to show a flexibility of choice of meta-training and meta-test device pool. We meta-train HELP with 6 devices, GPU Titan RTX, CPU Intel Xeon Gold, Mobile phone Pixel 2, Raspberry Pi, ASIC Eyeriss, and FPGA, then meta-test on 17 devices that are originally used as meta-training device. Pixel 3 is excluded from the test device since HW-NAS-Bench [43] does not provide the latency values of architecture blocks to build a layer-wise predictor. We reports the average values of Spearman's rank correlation coefficient on 17 devices in Table C.2. HELP consistently shows a high correlation coefficient value, even if we use much less number of meta-training device.

## C.2 Searched Architecture Visualization on MobileNetV3 Space

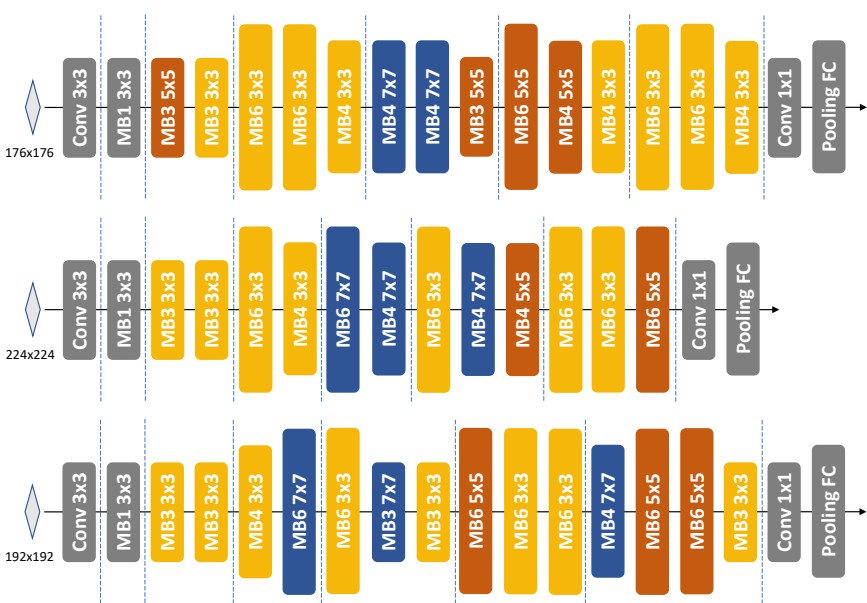

Figure C.1: Visualization of architectures searched by OFA+HELP on MobileNetV3 space. 20.3ms latency on Titan RTX GPU (batch size = 64) (top). 147ms latency on Xeon Gold 6226 CPU (batch size = 1) (middle). 67.4ms latency on Jetson AGX Xavier (batch size = 16) (bottom).

## C.3 Experiment on LatBench Dataset

**LatBench Dataset** is the latency dataset for various devices in the search space that is modified from the NAS-Bench-201 search space [31], provided by BRP-NAS [32]. Specifically, BRP-NAS simplifies

Table C.3: Performance comparison of FLOPs, BRP-NAS, and HELP on LatBench dataset, which has modified architecture space from NAS-Bench-201.

| Method | Samples from Target Devices | desktop cpu | mobile dsp 855 | mobile gpu 450 | Mean |
|:---:|:---:|:---:|:---:|:---:|:---:|
| FLOPs | - | 0.706 | 0.803 | 0.955 | 0.821 |
| BRP-NAS | 10 | 0.701 | 0.308 | 0.775 | 0.594 |
| BRP-NAS | 900 | 0.991 | 0.959 | 0.961 | **0.970** |
| HELP (Ours) | **10** | 0.990 | 0.958 | 0.956 | 0.968 |

the search space by removing "zero" and "skip-connect" operations from operation candidate set of NAS-Bench-201 to make consistency to NAS-Bench-101 [68]. Latencies are averaged over 1,000 latency samples while removing the lower and higher quartile values. Further, BRP-NAS removes 341 architectures that output zero in LatBench. On the other hand, when we construct our latency datasets, following HW-NAS-Bench [43], the full NAS-Bench-201 search is considered without any modification.

**Experimental Results** We validate the performance of our method on LatBench. The results in Table C.3 show that on the LatBench dataset, BRP-NAS indeed performs well, achieving high Spearman's correlation scores of 0.970 averaged on 3 test devices, and largely outperforming FLOPs. Since BRP-NAS achieves a high correlation score, HELP does not beat its performance, but still achieves a similar correlation score (0.968), using only 10 latency measurements, while BRP-NAS trained with 10 samples, on the other hand completely fails. Although HELP does not significantly outperform baseline on LatBench, our goal is not improving the accuracy of a device-specific latency predictor, but is in eliminating the main computational bottleneck of hardware-aware NAS, by proposing a latency predictor that is extremely sample-efficient and generalizes well to any hardware devices without requiring any knowledge of the target hardware devices.