# OpenReview forum: "Hardware-adaptive Efficient Latency Prediction for NAS via Meta-Learning"
_NeurIPS.cc/2021/Conference — NeurIPS 2021 Spotlight_

### Official Review · Reviewer_RCcn · 2021-06-30

**Rating:** 7
**Confidence:** 3

**Summary:**

Modern neural architecture search methods often require us to train a *latency prediction model* which can predict the latency of any candidate network architecture within a human-defined search space. These are commonly obtained by (i) sampling a set of random network architectures from the search space, (ii) benchmarking each one on the target inference hardware device, and (iii) training a regression model on these <architecture, latency> pairs in order to generalize to new, unseen architectures. The submission explores the idea of (i) training a single model which can generate predictions for many different hardware devices, and (ii) using meta-learning to allow the model to generalize to a new hardware device with little additional data (e.g., 10 samples).

**Limitations And Societal Impact:**

I believe the authors' discussion of limitations and societal impact in Appendix D adequately addresses the NeurIPS review requirements.

I would be interested in seeing a more detailed study of the meta-model's generalization performance within vs. across hardware families, but I think the submission already has enough interesting results for me to recommend its acceptance.

**Main Review:**

**Note**: I'll focus on the Neural Architecture Search / latency aspects of the submission (rather than the meta-learning aspects), since that's what I'm most familiar with.

# Overview
The paper's central claim is that meta-learning can be used to train a latency prediction model which can generalize to new (unseen) hardware devices with minimal training data (e.g., benchmarks of 10 random architectures from a large search space). The main steps are to (i) select a "seed" set of hardware devices, (ii) collect benchmark latencies for ~900 - 4000 random architectures for each device, and (iii) use the resulting <architecture, device, latency> triples to train a model which can predict latencies for architectures and hardware devices not seen in the training dataset (Section 4).

One of the inputs to the model is a representation of the target hardware device. This is generated in an interesting/clever way. The authors propose sampling/fixing 10 random architectures from the search space, and using the vector `<latency of architecture 1, latency of architecture 2, ...>` as their representation of the hardware device (Section 3.2). The submission proposes a "z-modulator" (weight adaptation mechanism) to adjust the network weights for a target hardware device.

The submission provides two sets of evaluations to test the proposed method. First (Figure 3, Figure 4, Figure 5, Table 2): evaluating how well the method performs on new hardware devices that were not seen during training, and comparing against a set of standard baselines (e.g. using FLOPS instead of on-device performance, comparing against a previously proposed method called BRP-NAS which trains a separate latency predictor for each device). In the authors' experiments, the newly proposed method significantly outperforms these baselines. And second (Table 3, Table 4, Table 5, Figure 6): plugging the proposed latency estimator into previously proposed NAS algorithms (e.g., Once-for-All, MetaD2A) and measuring the end-to-end quality impact on NASBench-201 or the Once-for-All search space. Both sets of experiments provide evidence that the latency predictor, once trained, can generalize reasonably well to new hardware devices with only ~10 benchmark samples.

**Clarity**: The submission seems clear and well-written.

**Significance**: The problem of trying to reduce the cost of training latency models for new hardware-devices seem well-motivated, and this is the first paper I've seen which tries to directly address this problem. It also seems like a nice application of ideas from few-shot learning. The results might be useful for practitioners who wish to simultaneously target a large number of different hardware devices.

**Incorrect Use of Big-O Notation**: The submission makes the following claim on Line 209 and Table 1 ("cost of NAS and latency estimation"):  "Since conventional approaches require to collect a large number of samples and train a device-specific latency estimator for each target device, while HELP only needs to collect 10 samples per device, HELP reduces the time complexity of obtaining latency estimations on N devices from O(N) to O(1)."

This seems like an incorrect use of big-O notation. The cost of training models for N hardware devices is reduced from `900*N` to `10*N+C`, where C is the cost of training the meta-model. Both of these costs are technically `O(N)`, since the cost grows linearly with the number of target devices as N goes to infinity. **Because the claims about big-O complexity are technically incorrect, I would not feel comfortable accepting the paper until the issue is addressed, or until the authors clarify their meaning (if I misunderstood their claims).**

**Caveats (different devices in same family may be strongly correlated)**: While the ability to adapt to a new hardware device with only 10 samples is impressive, it's possible that this is possible because many of the devices in the submission's pool are quite similar to each other. For example, it might be reasonable to expect Intel Xeon Silver and Intel Xeon Gold CPUs to have strongly correlated performance. This is significant because (i) it's not obvious to me that we actually need separate neural network architectures for the two CPUs, and (ii) it's possible that we could get an excellent predictor for Intel Xeon Gold by taking latency predictions for Xeon Silver and rescaling them a bit, without any kind of fancy meta-learning tricks.

Overall, the target devices the authors evaluate can be broken down into the following categories (Appendix A.2):
1. Desktop GPU (5 devices, all made by NVIDIA).
2. Edge GPU (1 device, made by NVIDIA).
3. Server CPU (3 devices, all made by Intel).
4. Mobile Phone CPU (5 devices, made by Samsung and Qualcomm)
5. Raspberry Pi (1 device).
6. ASIC (1 device, made by Eyeriss).
7. FPGA (1 device).

I have mixed feelings about this dataset. On the positive side, it covers a diverse set of hardware devices of different types/sizes. On the negative side, if we split the list of devices randomly, we might end up with an overly optimistic estimate of the model's performance. Given a random split, with high probability, at least one mobile desktop GPU will end up in the training set and another very similar device will end up in the test set.

For this reason, I think that evaluating the latency predictor's ability to generalize across device families is importance. The submission takes a first step towards addressing this in Table 5, where the meta-model is trained only on CPUs but is evaluated on GPUs + Intel CPUs. This suggests that in terms of end-to-end performance (plugging in the latency predictor to a previously proposed NAS algorithm), the current method gets reasonable generalization across hardware families. But it would be helpful to see a more careful analysis of the predictor's raw performance as well.

# Additional Detailed Comments
Figure 3: Each line shows a shaded error region, but I didn't notice an explanation for how these shaded regions were derived (e.g., worst run vs. 1 standard deviation below the mean).

**Time Spent Reviewing:**

5

---

> ### Author Response · Authors · 2021-08-10
> **Response to reviewer RCcn**
>
> **(1) Incorrect use of Big-O notation.**
>
> - As Reviewer RCcn pointed out, we agree that the computational complexities reported here in big O notations are misleading. We will revise the O(1) cost by HELP to O(D), and baselines by O(DN), where D is the number of devices and N is the number of samples to sufficiently train each latency estimator in Table 1. We thank the reviewer for the correction.
>
> ***
> **(2) Caveats (different devices in same family may be strongly correlated).** While the ability to adapt to a new hardware device with only 10 samples is impressive, it's possible that this is possible because many of the devices in the submission's pool are quite similar to each other. For example, it might be reasonable to expect Intel Xeon Silver and Intel Xeon Gold CPUs to have strongly correlated performance. This is significant because (i) it's not obvious to me that we actually need separate neural network architectures for the two CPUs, and (ii) it's possible that we could get an excellent predictor for Intel Xeon Gold by taking latency predictions for Xeon Silver and rescaling them a bit, without any kind of fancy meta-learning tricks.
>
> - We appreciate your insightful and thoughtful comments. Different devices from the same hardware platform **can have very different latencies on the same set of neural architectures**. In the table below, we report Spearman’s rank correlation coefficient between measured ground truth latencies of multiple hardware devices from each hardware platform on the same set of architectures. The results show that the measured latencies on the same set of architectures are **largely different** (e.g., pixel 2 vs pixel 3, GPU 1080ti vs 2080ti with batch size 256). Furthermore, even with the same GPU device, the **correlation scores are not that high** if the batch sizes are different. Thus, we cannot simply use a latency estimator from a device for a latency estimation of another device from the same platform, and expect it to have high accuracy. Therefore, including devices of the same platform in the training pool does not guarantee an overly optimistic performance on the target unseen device.
>
> -  As the reviewer pointed out, there are some cases where devices are relatively correlated (e.g., CPU silver 4114 vs gold 6226), yet HELP can raise the correlation coefficient from 0.91 to 0.98 of gold 6226 which is not negligible in HW-NAS process. We believe that random split is fair enough to demonstrate our model’s ability to adapt to an unseen device.
>
> |             	|          	|            	|          	|            	|             	|           	|          	|         	|
> |-------------	|----------	|------------	|----------	|------------	|-------------	|-----------	|----------	|---------	|
> |             	| 1080ti 1 	| 1080ti 256 	| 2080ti 1 	| 2080ti 256 	| silver 4114 	| gold 6226 	| pixel 2  	| pixel 3 	|
> |   1080ti 1  	|     1    	|    0.74    	|   0.96   	|    0.78    	|     0.83    	|    0.96   	|   0.80   	|   0.59  	|
> |  1080ti 256 	|          	|      1     	|   0.72   	|    0.82    	|     0.76    	|    0.78   	|   0.77   	|   0.74  	|
> |   2080ti 1  	|          	|            	|     1    	|    0.76    	|     0.81    	|    0.94   	|   0.79   	|   0.57  	|
> |  2080ti 256 	|          	|            	|          	|      1     	|     0.95    	|    0.88   	|   0.88   	|   0.87  	|
> | silver 4114 	|          	|            	|          	|            	|      1      	|    0.91   	|   0.87   	|   0.81  	|
> |  gold 6226  	|          	|            	|          	|            	|             	|     1     	|   0.86   	|   0.71  	|
> |   pixel 2   	|          	|            	|          	|            	|             	|           	|     1    	|   0.76  	|
> |   pixel 3   	|          	|            	|          	|            	|             	|           	|          	|    1    	|
> ***
> **(3) Figure 3: Each line shows a shaded error region, but I didn't notice an explanation for how these shaded regions were derived (e.g., worst run vs. 1 standard deviation below the mean).**
> - Shaded regions are mean+/- standard deviation of 5 runs with random seeds. We will add this to the paper for clarity.
>
> ***
> We thank you for your kind suggestions for our paper and will include the above discussions and corrections in the revision.

---

> > ### Comment · Reviewer_RCcn · 2021-08-18
> > **Thanks for your feedback**
> >
> > The big-O analysis correction that you mentioned looks good. It should address the concerns I raised in my original review.
> >
> > The correlation table you provided in your response is quite interesting, and partially addresses my concern about similarity between different devices in the same family. One caveat is that the current table does not take measurement error into account. For example: the correlation between any device and itself is listed as 1 in the table. But if you benchmark the same device twice, you'll get two slightly different sets of numbers. Without this baseline, it's unclear to me how much of the difference between devices in the same family is due to measurement error vs. intrinsic hardware design differences. Probably the cleanest way to address this concern would be to re-benchmark each device and then replace each "1" in the table with the correlation between the first and second set of benchmark results.
> >
> > For now, I've left my original score (7) unchanged, and will continue to recommend the paper's acceptance.

---

> > > ### Author Response · Authors · 2021-08-24
> > > **Additional response to reviewer RCcn**
> > >
> > > **Comment** The big-O analysis correction that you mentioned looks good. It should address the concerns I raised in my original review.
> > >
> > > - We are happy to address the reviewer’s concern. We will update the paper with this correction version.
> > >
> > > **Comment** The correlation table you provided in your response is quite interesting, and partially addresses my concern about similarity between different devices in the same family. One caveat is that the current table does not take measurement error into account. For example: the correlation between any device and itself is listed as 1 in the table. But if you benchmark the same device twice, you'll get two slightly different sets of numbers. Without this baseline, it's unclear to me how much of the difference between devices in the same family is due to measurement error vs. intrinsic hardware design differences. Probably the cleanest way to address this concern would be to re-benchmark each device and then replace each "1" in the table with the correlation between the first and second set of benchmark results.
> > >
> > > - Following the reviewer’s comments, we re-measure and rank latency values of architectures for each device. Then we compute the correlation between the first and second set of each device and report the results as shown in the below Table.
> > > |             	| 1080ti 1 	| 1080ti 256 	| 2080ti 1 	| 2080ti 256 	| silver 4114 	| gold 6226 	| pixel 2  	| pixel 3 	|
> > > |:-----------:	|:--------:	|:----------:	|:--------:	|:----------:	|:-----------:	|:---------:	|:--------:	|:-------:	|
> > > |   1080ti 1  	|   1.00   	|    0.74    	|   0.96   	|    0.78    	|     0.83    	|    0.96   	|   0.80   	|   0.59  	|
> > > |  1080ti 256 	|          	|    1.00    	|   0.72   	|    0.82    	|     0.76    	|    0.78   	|   0.77   	|   0.74  	|
> > > |   2080ti 1  	|          	|            	|   0.99   	|    0.76    	|     0.81    	|    0.94   	|   0.79   	|   0.57  	|
> > > |  2080ti 256 	|          	|            	|          	|    1.00    	|     0.95    	|    0.88   	|   0.88   	|   0.87  	|
> > > | silver 4114 	|          	|            	|          	|            	|     0.98    	|    0.91   	|   0.87   	|   0.81  	|
> > > |  gold 6226  	|          	|            	|          	|            	|             	|    0.97   	|   0.86   	|   0.71  	|
> > > |   pixel 2   	|          	|            	|          	|            	|             	|           	|   0.94   	|   0.76  	|
> > > |   pixel 3   	|          	|            	|          	|            	|             	|           	|          	|   1.00  	|
> > > - We believe that correlation values for the device itself become more accurate by reflecting measurement error. We will include this version Table in the manuscript. Thank you for the insightful suggestions.

---

### Official Review · Reviewer_FZ6V · 2021-07-14

**Rating:** 7
**Confidence:** 5

**Summary:**

This paper presents a hardware latency modeling and prediction methodology that adapts itself to multiple devices through few shot meta-learning. The overall goal of the paper is to create a hardware predictor for use within hardware-aware neural architecture search (NAS).

A graph convolutional network (GCN) is first trained using latency measurements from a number of devices (18) then few-shot learning (with as little as 10 measurements) is done to adapt the predictor to predict the latency of unseed devices (3 devices are similar to the devices used for training, and 3 devices are different). The problem addressed with this work is important as it is often impractical to get numerous measurements from real hardware devices. Few-shot metalearning is one way to adapt a latency predictor for a new device and this is the first time I have seen metalearning applied to the specific problem of hardware latency for different devices.


**Limitations And Societal Impact:**

The main limitations of the paper are stated in the main review. I do not have comments on the societal impact as this work is quite technical and abstracted away from the important societal issues in the machine learning community.


**Main Review:**

While the results in the paper are strongly in favour of the proposed methodology compared to prior work, there are many questionable assumptions in those comparisons and missing ablations that need to be addressed to prove the efficacy of the proposed meta-learning approach. I am strongly in favour of seeing this work published but I find it hard to give a high score without addressing the major issues listed below.

### Strengths:
The topic is important and interesting and the use of meta-learning is spot-on. This is a great way to formulate the hardware latency prediction problem and provide a low-cost way to adapt to different devices.

### Major issues:
- Without code release, it is difficult to trust the results.
- Choice of "unseen device" and "unseen platform" seems to be hand-picked for good results. An ablation would really help in understanding the pros/cons of using HELP. For example, what happens with fewer train devices? What happens when you swap train and test devices? What happens when your "completely unseen platforms" more complex? Latency prediction for ASIC, FPGA and MCU devices is *very* simple and could usually be modeled analytically by counting the number of compute cycles. Additionally, the "unseed devices" CPU/GPU/Pixel are **very** similar to devices in the meta-train pool for which many training samples are provided.
- Some results are overstated. When comparing the number of samples used to train each predictor, all training points should be reported in some way. For HELP in Table 3 for example, it only has '10' measurements needed but the actual number is 900*18+10=16210. I do understand that most of this is a fixed cost that is not incurred with each new device but it still needs to be clearly spelled out -- I found the current presentation of results slightly misleading in some places.
- Again in Table 3, it is difficult to discern **why** HELP is better than BRP-NAS? Both predictors are based on GCNs. The only reason I can think of has to be related to the fact that HELP is trained on more data points (18 devices * 900 points) and so that data allows it to achieve a better prediction. I would've expected HELP to be an *approximation* of BRP-NAS so to speak. A discussion of the reason behind this improvement is warranted.

### Minor issues:
- Equation 3 is odd. It combines set notation and function notation. It is unclear to me what it is saying.
- Why are some results missing? For example in Table 3 the layerwise predictor is not present for 4 devices.
- In Table 3, why is the FLOPs correlation coefficient so high for the GPU when Fig. 4(a) suggests that it should be very low?
- I don't view this as the main contribution of the work, but in the comparison to baseline metalearning approaches like meta-SGD in Fig. 5, the baseline seem to be completely failing - is it just untuned? or is this a trivial baseline? Or is the fact that you use a modulator making all the difference? A discussion on this would help.
- Table 1 is misleading in my opinion.

## Post-rebuttal
All of my concerns were addressed - please see the conversation below with the authors. I will increase my score to reflect that I am satisfied with their responses. As mentioned below: With the code release and additional ablations (especially ones that show that this method generalizes with different devices in test vs. train pool), I would like to vote for accepting this paper. I think it is a solid contribution that straddles the boundary between two important areas: metalearning and hardware device modeling. I would suggest adding many of the ablations in this rebuttal to the submitted paper (or its appendix).

**Time Spent Reviewing:**

4

---

> ### Author Response · Authors · 2021-08-09
> **Response to reviewer FZ6V - Part 2 (2/2)**
>
> **(4) Again in Table 3, it is difficult to discern why HELP is better than BRP-NAS? Both predictors are based on GCNs. The only reason I can think of has to be related to the fact that HELP is trained on more data points (18 devices * 900 points) and so that data allows it to achieve a better prediction. I would've expected HELP to be an approximation of BRP-NAS so to speak. A discussion of the reason behind this improvement is warranted.**
> - This is a misunderstanding. Simply training a single latency estimator on latency measurements from multiple devices will not work. This is because we need a way to **adapt**the latency estimator to each device, which in turn requires to **represent**each hardware device. These are nontrivial problems, and since BRP-NAS does not have a mechanism to modify the latency estimator for each device, training it on multiple heterogeneous devices will only make the prediction to simply predict the **average latency**across all devices, for a given architecture.
>
> - Further, simply meta-learning the latency estimator with conventional methods is highly suboptimal, as shown with the case of Meta-SGD in Figure 5. The latency estimator trained with Meta-SGD, is trained with the exactly the same number of data points from 18 devices, but has a very low performance (around 0.7 correlation score at best), although it can “adapt” to a new device by taking gradient steps with the samples collected from the target device.
>
> - The main contribution of our work is the proposal of a meta-latency predictor that can **adapt**to a new device, by making it as a **function of the device**. However, to model the latency estimator as a function of the device, we need to **represent**each device. To solve this challenging problem, we propose a novel device embedding which represents each device by the measured latencies of it on a small number of reference architectures. Then, we perform amortized meta-learning based on this device embedding, to make the latency estimator as a function of the device. Finally, we use a modulator z based on the device embedding (Eq. 6) to further modify the parameter of the latency estimator in a device adaptive manner. These are very important, and non-straightforward technical contributions we make, which maximizes the adaptation performance of the meta-latency predictor.
>
> - To further show the above points, we trained BRP-NAS and HELP on the same number of latency measurement samples from 18 devices, then reported their Spearmans' rank correlation coefficients on 6 meta-test devices. As shown in the table below, we observe that HELP largely outperforms BRP-NAS trained with 900*18 pre-training samples, which only achieves negligible performance gain over BRP-NAS without pre-training steps.
>
> |    Method   	| Pre-Training Samples 	| Samples From Target 	|  GPU  	|  CPU  	| Pixel2 	| Raspi4 	|  ASIC 	|  FPGA 	| Mean Correlation 	|
> |:-----------:	|:--------------------:	|:-------------------:	|:-----:	|:-----:	|:------:	|:------:	|:-----:	|:-----:	|:----------------:	|
> |   BRP-NAS   	|           -          	|         900         	| 0.814 	| 0.796 	|  0.757 	|  0.847 	| 0.811 	| 0.801 	|       0.804      	|
> |   BRP-NAS   	|        900*18        	|         900         	| 0.808 	| 0.783 	|  0.780 	|  0.850 	| 0.819 	| 0.797 	|       0.806      	|
> | HELP (Ours) 	|        900*18        	|          10         	|  **0.996** | **0.983** |  **0.884** |  **0.873** | **0.971** | **0.987** | **0.949** |
>
> ***
> **(5) Equation 3 is odd. It combines set notation and function notation. It is unclear to me what it is saying.**
> - We appreciate your constructive comment. We will remove the function notation (E \times H -> R^d) in Equation 3 for clarity.
>
> ***
> **(6) Why are some results missing? For example, in Table 3 the layerwise predictor is not present for 4 devices.**
> - With a layer-wise predictor, the latency of a neural architecture is estimated by summing up the estimated latency for its individual blocks. For GPUs and CPUs we could directly measure and obtain layer-wise latency data. However, for the three other devices (Raspi4, ASIC, and FPGA), we only use the data provided by HW-NAS-Bench that only includes the latencies for each architecture, without any latency measurements for each layer. In addition, for Pixel 2, a correlation value with the layer-wise predictor is 0.834. We will add this value in the main paper. Thank the reviewer for the constructive comment.
>
> ***
> **(7) In Table 3, why is the FLOPs correlation coefficient so high for the GPU when Fig. 4(a) suggests that it should be very low?**
> - In Table 3, we report the Spearman’s rank correlation coefficient between the estimated latencies and the actual latencies, which is a standard metric to measure the performance of the latency estimation. It measures the statistical dependence between the **ranking** of the two variables, and not their absolute values. Thus, even with the plot like Fig. 4(a), the rank correlation coefficient can be high. On the other hand, HELP achieves very high performance either with Spearman’s rank correlation, or conventional correlation measure.
>
> ***
> **(8) I don't view this as the main contribution of the work, but in the comparison to baseline meta learning approaches like meta-SGD in Fig. 5, the baseline seem to be completely failing - is it just untuned? or is this a trivial baseline? Or is the fact that you use a modulator making all the difference? A discussion on this would help.**
>
> - Meta-SGD is not a trivial baseline but a very powerful meta-learning baseline, and it is fully tuned. The comparison between HELP and Meta-SGD in Fig. 5 actually emphasizes the main contribution of our work, as it shows that a naive application of conventional meta-learning algorithms for device-adaptive latency estimation does not work.
> - Meta-SGD, is a **hardware-agnostic**meta-learning baseline which meta-learns a shared parameter of a latency estimator, that can adapt well to any devices from the meta-training pool, by taking a few gradient steps with the latency samples from each device. Although this baseline is able to adapt to a new device by taking inner gradient steps with the new samples collected from it, this strategy is highly suboptimal since there is no way to discriminate between different devices, during the meta-training stage, or when adapting to a new device at the meta-test time.
>
> - HELP, on the other hand, is a **hardware-adaptive** meta-latency estimator that is trained as a function of the device, whose parameters can be further modulated with a mask obtained as a function of the device. However, in order to train a latency estimator as a function of the device, we need to represent devices with highly heterogeneous hardware platforms in a unified way, which is highly challenging. We tackle this with a clever hardware-device embedding which represents each device as a set of latency measurements on a set of reference architectures, that allows embedding of any hardware devices.
>
> - Thus, while HELP can discriminate between different devices during both meta-training and meta-test (device-adaptation) time, Meta-SGD fails to discriminate latency values of the same architecture measured on different devices during meta-training. This is why Meta-SGD has a low performance even when it can adapt to each device and is trained with the same number of meta-training samples. We have this discussion in the paragraph “Effect of the Hardware-adaptive Meta-learning of Section 4.1: Line 262-267), but we will include the above discussion for further clarification.
>
> ***
> **(9) Table 1 is misleading in my opinion.**
> - As Reviewer RCcn also pointed out, we agree that the computational complexities reported here in big O notations are misleading. We will revise the O(1) cost by HELP to O(D), and baselines by O(DN), where D is the number of devices and N is the number of samples to sufficiently train each latency estimator in Table 1. We thank the reviewers for the correction.
>
> We believe that we have faithfully addressed all your concerns, and politely ask you to revise the score and the review accordingly. We thank you again for your time and efforts in reviewing our paper.

---

> > ### Comment · Reviewer_FZ6V · 2021-08-09
> > **A convincing response!**
> >
> > Thank you for your detailed response to my questions. With the code release and additional ablations (especially ones that show that this method generalizes with different devices in test vs. train pool), I would like to vote for accepting this paper. I think it is a solid contribution that straddles the boundary between two important areas: metalearning and hardware device modeling. I would suggest adding many of the ablations posted here to the submitted paper (or its appendix).
> >
> > I did have two outstanding questions still that I am hoping that the authors will answer.
> >
> > 1. The first question is a minor one: when you say `conventional correlation` in the quote below, do you mean Pearson's correlation coefficient? It would be good to clearly define and state which correlation you use in each plot/table. The inconsistency that I pointed out is plausible if you use Pearson and Spearman in the table and plot respectively, but it was not clear in the submitted manuscript.
> >
> > > In Table 3, we report the Spearman’s rank correlation coefficient between the estimated latencies and the actual latencies, which is a standard metric to measure the performance of the latency estimation. It measures the statistical dependence between the ranking of the two variables, and not their absolute values. Thus, even with the plot like Fig. 4(a), the rank correlation coefficient can be high. On the other hand, HELP achieves very high performance either with Spearman’s rank correlation, or conventional correlation measure.
> >
> > 2. Regarding BRP-NAS. I am still unconvinced of is the poor performance of BRP-NAS versus HELP. They are both GCN-based predictors. How can a BRP-NAS predictor, trained on a single device with 900 measurements from that specific device still perform so badly? My specific question is: how do you train BRP-NAS when predicting the latency on the Raspberry Pi for instance? Do you train it on 900 measurements from Rasberry Pi? Then when you evaluate on a different device, say Pixel 2 for instance, do you train a new BRP-NAS GCN on 900 points from Pixel 2? The text below is a bit confusing. I was not suggesting that you pre-train BRP-NAS on all devices because, by construction, it is a single-device latency estimator.
> >
> > > Simply training a single latency estimator on latency measurements from multiple devices will not work. This is because we need a way to adaptthe latency estimator to each device, which in turn requires to representeach hardware device. These are nontrivial problems, and since BRP-NAS does not have a mechanism to modify the latency estimator for each device, training it on multiple heterogeneous devices will only make the prediction to simply predict the average latencyacross all devices, for a given architecture.
> >
> > If my assumptions about BRP-NAS training above are correct, then I am still unconvinced of the accuracy gap to HELP. At least it is unintuitive to me that a device-specific latency predictor would underperform, by a large margin, compared to a general multi-device predictor. A further comment from the authors would help.

---

> > > ### Author Response · Authors · 2021-08-10
> > > **Additional response to reviewer FZ6V**
> > >
> > > **Comment 1.** Thank you for your detailed response to my questions. With the code release and additional ablations (especially ones that show that this method generalizes with different devices in test vs. train pool), I would like to vote for accepting this paper. I think it is a solid contribution that straddles the boundary between two important areas: meta-learning and hardware device modeling. I would suggest adding many of the ablations posted here to the submitted paper (or its appendix).
> > >
> > > **Answer 1.** We appreciate your positive feedback, and will include the discussions and new ablations in the revised version of the paper.
> > >
> > > ***
> > > **Comment 2.** I did have two outstanding questions still that I am hoping that the authors will answer. The first question is a minor one: when you say conventional correlation in the quote below, do you mean Pearson's correlation coefficient? It would be good to clearly define and state which correlation you use in each plot/table. The inconsistency that I pointed out is plausible if you use Pearson and Spearman in the table and plot respectively, but it was not clear in the submitted manuscript.
> > >
> > > **Answer 2.** We apologize for the confusion. We did not use different correlation coefficients, and reported Spearman’s rank correlation coefficients for all results in the paper, as described in Line 249 and 286 of the paper. What we wanted to say, regarding Figure 4, is that based on the results of the raw scatter plots of the predicted and measured latencies in Figure 4, ours will achieve significantly superior correlation scores regardless of whether we use Spearman’s rank correlation coefficients or Pearson’s correlation coefficients as the metric.
> > >
> > > ***
> > > **Comment 3.** Regarding BRP-NAS. I am still unconvinced of the poor performance of BRP-NAS versus HELP. How can a BRP-NAS predictor, trained on a single device with 900 measurements from that specific device still perform so badly?
> > >
> > > **Answer 3.**  We ran the official code from the authors and followed all their experimental settings, and thoroughly checked that all results are correct. Moreover, **BRP-NAS cannot be said to be working poorly**, since its average Spearman’s rank correlation score of 0.804 (with 900 samples) or 0.822 (with 3,200 samples) on 6 devices is high enough, as this is higher than the performance of the layerwise predictor and the FLOP based approximation (Please see Table 3).
> > >
> > > Thus the performance gap is not coming from the poor performance of the BRP-NAS, but is rather due to the effectiveness of the **device-adaptive meta-learning of HELP**. We will discuss this in the answer to the next comment.
> > >
> > > ***
> > > **Comment 4.** They are both GCN-based predictors. I am still unconvinced of the accuracy gap to HELP. At least it is unintuitive to me that a device-specific latency predictor would underperform, by a large margin, compared to a general multi-device predictor. A further comment from the authors would help.
> > >
> > > **Answer 4.**
> > > - Please note that the architecture is not the only factor that affects the performance, and **how we meta-learn the latency predictor** is the main reason of HELP’s success. Many meta-learned models have been shown to outperform the models trained with standard learning with exactly the same architecture, although they are trained on exactly the same set of training examples given for a novel task, since they can utilize meta-knowledge from the meta-training tasks.
> > >
> > > - Further, the performance can largely vary across meta-learned models, depending on which **meta-learning frameworks** they use. As shown in Table 2 and Figure 5, our HELP largely outperforms all meta-latency predictors (MAML, ANP, Meta-SGD) which are trained with **exactly the same set of meta-training tasks**. This performance gain mostly comes from our novel **device-embedding** and the **device-adaptive meta-learning** that is made possible with three meta-learning components, as shown in the ablation study in Table 2.
> > >
> > > - We summarize the differences between HELP and BRP-NAS below, focusing on each of its device-adaptive meta-learning components.
> > >
> > > 1)  HELP is trained as a function of the device with the help of novel device embedding we propose, and thus we can train a single latency estimator across multiple devices, while the BRP-NAS predictor requires training of a separate latency estimator for each device. This allows HELP to effectively transfer knowledge across heterogeneous devices, while the BRP-NAS predictor does not have such an ability (nor any meta-latency predictors trained with conventional meta-learning in Table 2).
> > >
> > > 2) HELP is meta-learned with bi-level optimization, such that the model can adapt to any new device or a task with few gradient steps. This means that HELP starts from a much better initialization than the BRP-NAS predictor, with the knowledge of the devices from the meta-training pool.
> > >
> > > 3) HELP utilizes a device-specific modulator z to transform the initial parameters for better latency estimation on the target device, which further allows the model to start from a better initialization than the BRP-NAS predictor, or HELP without the modulator.
> > >
> > >
> > > Each of these three hardware-adaptive meta-learning components are essential for HELP to achieve a very high score, and without them, the correlation score obtainable by HELP becomes very low (0.539), as shown in Table 2, which ablates the effectiveness of each component. Please note that all baselines in Table 2 use exactly the same GCN-based predictors, but their performances largely vary.
> > >
> > > ***
> > > **Comment 5.** My specific question is: how do you train BRP-NAS when predicting the latency on the Raspberry Pi for instance? Do you train it on 900 measurements from Rasberry Pi? Then when you evaluate on a different device, say Pixel 2 for instance, do you train a new BRP-NAS GCN on 900 points from Pixel 2?
> > >
> > > **Answer 5.**  Yes. This is what we do because the BRP-NAS predictor is a single-device latency estimator.

---

> ### Author Response · Authors · 2021-08-09
> **Response to reviewer FZ6V - Part 1 (1/2)**
>
>
> **(1) Without code release, it is difficult to trust the results.**
> - We did release the code and the dataset for reproduction, in Section E of the supplementary file (https://anonymous.4open.science/r/HELP-385D/README.md). We will move the link to the code to the first page of the main paper.
>
> ---
>
> **(2-1) Choice of “unseen device” and “unseen platform” seems to be hand-picked for good results.**
> - This is not true. We did not hand-pick the devices or platforms for good results, and simply constructed the meta-training and meta-test pool to have a wide variety of devices.
>
> ---
>
> **(2-2) An ablation would really help in understanding pros/cons of using HELP.**
> - We already have ablation studies that show how HELP works with fewer training/test devices in the main paper, and you may have missed them. We provide the details of them below.
>
> ---
>
> **(2-3) What happens with fewer train devices?**
> - We have an entire paragraph titled “Effect of the Size of the Device Pool”, in Section 4.1 (Line 268-276), and Figure 5 shows how the performance changes with the different size of the devices used for meta-training. As shown in Figure 5 and mentioned in Line 272, when the number of devices in the meta-training pool is **10 or more**, the predicted latency achieves a very high correlation score with the actual latencies measured on unseen devices from 6 different platforms, that is close to the score obtained with 18 devices in the meta-training pool. HELP also obtains only slightly lower correlation score with a meta-training pool which only consists of **4 devices**, compared to the model meta-trained with 18 devices. However, when using only two devices for meta-training, then HELP achieves a similar performance to Meta-SGD since the effectiveness of the hardware-adaptive meta-learning diminishes in such a case.
>
> ---
>
> **(2-4) What happens when you swap train and test devices?**
> - Thank you for the helpful suggestion. Following your comment, we trained the latency estimators on 6 devices that are originally used as the meta-test devices, and then computed the correlation of the estimated latency using the meta-latency predictor on 18 unseen devices that are originally used as the meta-training devices in Table 3 of the original paper. The Table below shows the average values of Spearman's rank correlation coefficient. (For Layer-wise Predictor, we compute an average value on 17 unseen devices except Pixel3 provided by HW-NAS-Bench since they do not provide latency values of architecture blocks to build layer-wise predictor.)
>
> |                             	| FLOPS 	| Layer-wise Predictor 	| BRP-NAS 	| BRP-NAS (+extra examples) 	| HELP (Ours) 	|
> |:---------------------------:	|:-----:	|:--------------------:	|:-------:	|:-------------------------:	|:-----------:	|
> | Samples From Target Devices 	|   -   	|           -          	|   900   	|            3200           	|      **10**     	|
> |      Mean Corr. Coeff.      	| 0.747 	|         0.858        	|  0.773  	|           0.793           	|    **0.940**    	|
>
> - We observe that although we use much less number of meta-training devices (6) than in the original paper (18), our model, HELP consistently shows high correlation coefficient values on 18 unseen devices (average: 0.940).
>
> ---
>
> **(2-5) What happens when your "completely unseen platforms" more complex? Latency prediction for ASIC, FPGA and MCU devices is very simple and could usually be modeled analytically by counting the number of compute cycles.**
> - First of all, to our knowledge, our work considers the most extensive set of hardware platforms compared to existing works on hardware-aware NAS, as shown in the table below:
>
> |                 	| Desktop GPU 	| Server or Desktop CPU 	| Mobile Phone 	| Edge GPU 	| FPGA 	| Edge TPU 	| ASIC  	| Raspberry Pi 	|
> |:---------------:	|:-----------:	|:---------------------:	|:------------:	|:--------:	|:----:	|:--------:	|:-----:	|:------------:	|
> |      OFA[1]     	|      o      	|           o           	|       o      	|     o    	|   o  	|          	|       	|              	|
> |    BRP-NAS[2]   	|      o      	|           o           	|       o      	|     o    	|      	|     o    	|       	|              	|
> |    CompOFA[3]   	|      o      	|           o           	|       o      	|          	|      	|          	|       	|              	|
> | HardCoRe-NAS[4] 	|      o      	|           o           	|              	|          	|      	|          	|       	|              	|
> |   HELP (Ours)   	|      o      	|           o           	|       o      	|     o    	|   o  	|         	|   o   	|       o      	|
>
> [1] OFA: Cai, Han, et al. "Once-for-all: Train one network and specialize it for efficient deployment", ICLR 2020
>
> [2] BRP-NAS: Dudziak, Łukasz, et al. "BRP-NAS: Prediction-based NAS using GCNs", NeurIPS 2020
>
> [3] CompOFA: Sahni, Manas, et al. "Compound Once-For-All Networks for Faster Multi-Platform Deployment", ICLR 2021
>
> [4] HardCoRe-NAS: Nayman, Niv, et al. "Hard Constrained diffeRentiable Neural Architecture Search", ICML 2021
>
> ---
>
> **(2-6) Additionally, the "unseen devices" CPU/GPU/Pixel are very similar to devices in the meta-train pool for which many training samples are provided.**
>
> - CPUs, GPUs, and mobile devices are the most general set of hardware platforms and thus having them in the meta-training pool is a natural choice. Also, we do not find the hardware devices to be that similar. You seem to have missed the part that our method is able to adapt the latency estimator to a completely new device from an unseen platform, as 1) it gets the device embedding of the novel device as an input (Eq. 4), and 2) further modulates the parameter of the latency estimator via the mask z (Eq. 6).
>
> - Moreover, different devices from the same hardware platform **can have very different latencies on the same set of neural architectures**. In the table below, we report Spearman’s rank correlation coefficient between measured ground truth latencies of multiple hardware devices from each hardware platform on the same set of architectures. The results show that the measured latencies on the same set of architectures are **largely different** (e.g., pixel 2 vs pixel 3, GPU 1080ti vs 2080ti with batch size 256). Furthermore, even with the same GPU device, the **correlation scores are not that high** if the batch sizes are different. Thus, we cannot simply use a latency estimator from a device for a latency estimation of another device from the same platform, and expect it to have high accuracy.
>
> |             	|          	|            	|          	|            	|             	|           	|          	|         	|
> |-------------	|----------	|------------	|----------	|------------	|-------------	|-----------	|----------	|---------	|
> |             	| 1080ti 1 	| 1080ti 256 	| 2080ti 1 	| 2080ti 256 	| silver 4114 	| gold 6226 	| pixel 2  	| pixel 3 	|
> |   1080ti 1  	|     1    	|    0.74    	|   0.96   	|    0.78    	|     0.83    	|    0.96   	|   0.80   	|   0.59  	|
> |  1080ti 256 	|          	|      1     	|   0.72   	|    0.82    	|     0.76    	|    0.78   	|   0.77   	|   0.74  	|
> |   2080ti 1  	|          	|            	|     1    	|    0.76    	|     0.81    	|    0.94   	|   0.79   	|   0.57  	|
> |  2080ti 256 	|          	|            	|          	|      1     	|     0.95    	|    0.88   	|   0.88   	|   0.87  	|
> | silver 4114 	|          	|            	|          	|            	|      1      	|    0.91   	|   0.87   	|   0.81  	|
> |  gold 6226  	|          	|            	|          	|            	|             	|     1     	|   0.86   	|   0.71  	|
> |   pixel 2   	|          	|            	|          	|            	|             	|           	|     1    	|   0.76  	|
> |   pixel 3   	|          	|            	|          	|            	|             	|           	|          	|    1    	|
>
> ---
>
> **(3) Some results are overstated. When comparing the number of samples used to train each predictor, all training points should be reported in some way. For HELP in Table 3 for example, it only has '10' measurements needed but the actual number is 900*18+10=16210. I do understand that most of this is a fixed cost that is not incurred with each new device but it still needs to be clearly spelled out -- I found the current presentation of results slightly misleading in some places.**
> - We believe that only counting the number of training samples from the target device is fair, since none of the baselines can utilize samples from the other devices, even if they are available. We will discuss more about this point in detail in the response to the next comment. However, as suggested, we will revise “Sample” in Table 3 and 4 to “Samples from the Target Device”, and will include a description that “HELP is meta-learned with 900*18 data samples from *other devices* in the meta-training pool” in Section 4.1 and 4.2.

---

> ### Author Response · Authors · 2021-08-11
> **We sincerely appreciate your time and effort in reviewing our paper.**
>
> We are happy to clearly address all of Reviewer FZ6V’s concerns. We will include the Reviewer FZ6V’s suggestions to improve our paper, especially, the experimental results to demonstrate the generalization of HELP on more devices (18 devices from meta-training pool), in the revision. We believe these things make our paper stronger and thank the Reviewer FZ6V again for the active discussions and thoughtful comments.

---

### Official Review · Reviewer_rNFs · 2021-07-16

**Rating:** 7
**Confidence:** 4

**Summary:**

This paper aims to improve the estimation of latency of neural models on unseen hardware devices. Its contributions are:
- Utilize a collected pool of devices and architectures to meta-train the predictor.
- Adapt the predictor to unseen hardware devices/platforms by considering hardware device embedding -- the latencies of the unseen devices on a fixed set of reference neural architectures.
- Introduce a "hardware-adaptive modulator" to improve the prediction.
- Reduce the latency estimator building cost of N devices from O(N) to O(1).
- Apply the latency estimator to latency-constrained NAS.

**Limitations And Societal Impact:**

The authors have addressed the limitations and potential negative societal impact of their work.

**Main Review:**

**Originality**
The contributions of this paper are quite outstanding:
- Improve SOTA latency predictor by formulating it as a few-shot regression problem.
- Use latencies on reference architectures as hardware embeddings.
- Combine with exisiting NAS framework to accelerate latency-constrained NAS.
- Good combination of existing techniques: GCN and MLP to encode architectures and devices, meta learning to adapt the predictor to unseen devices.
- Improve number of samples required to adapt to an unseen device.

**Quality**
- The proposed methods are sound and convincing.
- The empirical results support the arguments.
- The meta-learning approach looks simple and effective.
- The hardware-adaptive modulator needs more explanation.

**Clarity**
Some explanations in the paper are not clear. I hope the author can address the problems and make it a solid paper.
- Table 2: what are HW-condition, few-shot adapt and z modulator? It will be clearer to relate to the sections explicitly.
- Hardware-adaptive modulator seems to be an important contribution. However the explanation is not very clear -- How is the hardware-adaptive modulator be designed, implementated and trained?
- Why is the latency model builing time so long for ProxylessNAS and OFA-layerwise? They use layer-wise predictor which are the summation of profiled per-layer latency. Should be fast.
- Check mathematical notations. For example: f vs f^{\tau}, V_h^{(0)} -- what is (0)? 2)
- Figure 3 and 5: What is the unseen device? How to select the device in the pool for each point in the x-axis?

**Significance**
- The paper has reduced the latency estimator building cost by meta-learning a few-shot regression model that can generalize across hardware devices.
- The source code is available to reproduce the results. It would be more helpful if the dataset (latency measurement from the device pool) is also available.

**Post-rebuttal updates**
Thank you to the authors for answering my questions and clarifying the technical details. I am convinced that meta learning + hardware performance predictor is a solid contribution. I will increase my score to accept this paper.

**Time Spent Reviewing:**

3

---

> ### Author Response · Authors · 2021-08-10
> **Response to reviewer rNFs**
>
> **(1) Table 2: what are HW-condition, few-shot adapt and z modulator? It will be clearer to relate to the sections explicitly.**
>
> - Thank you for the helpful suggestion. For clarity, we will add descriptions for each component in the caption and the text. ‘HW-condition’ refers to the hardware-conditioned latency prediction model in Equation (2) which gets the hardware-embedding encoded by Equation (3) in Section 3.2 as an input. ‘Few-shot adapt’ denotes that we perform inner gradient updates with the samples collected from the novel device, as described in Equation (5), in Section 3.3. ‘z modulator’ denotes the model which additionally performs hardware-specific modulation of the parameters of the meta-latency predictor, as described in Equation (6) and Equation (7) in Section 3.3.
>
> ***
> **(2) Hardware-adaptive modulator seems to be an important contribution. However the explanation is not very clear. How is the hardware-adaptive modulator be designed, implemented and trained?**
>
> - We provided the implementation and training details of our HELP, including the hardware-adaptive modulator in “Section B Implementation and Training Details” of the supplementary file, due to the page limit.
>
> - Specifically, the $z$ Modulator takes hardware device embedding as an input, and outputs the 603-dimensional scaling $z$ vector. This $z$-vector is element-wisely multiplied to the weights and the biases of the latency predictor, which is a three-layer MLP with 100-dimensional hidden layers, which receives the concatenation of the architecture embedding as inputs, and outputs the estimated latency as a scalar value. This modulation of the parameter of the meta-latency predictor will modify its latency-estimating behavior according to the given device.  Please check Section B of the supplementary file for more details.
>
> - However, we want to further emphasize that the hardware-adaptive modulator is not the only mechanism which allows the meta-latency estimator to adapt to each device. The device adaptation is done with three mechanisms: 1) By using the **device embedding**as an input for a latency estimation model, we can meta-train the latency estimator in a device adaptive manner, and also can adapt it to a novel device. 2) By taking gradient steps using the latency samples collected from a new device, we can adapt it to estimate the latency for a new device. 3) Utilizing the hardware-adaptive modulator further allows us to modify the latency estimator’s parameters without taking any gradient steps. We show that all three components are effective in improving the device-adaptive latency estimation performance, in the ablation study in Table 2.
>
> ***
> **(3) Why is the latency model building time so long for ProxylessNAS and OFA-layerwise? They use layer-wise predictor which are the summation of profiled per-layer latency. Should be fast.**
>
> - We did use the layer-wise predictor for ProxylessNAS and OFA-layerwise. The main time bottleneck in the process of building a latency prediction model is the time to collect **latency measurements**, to build a training set consisting of (architecture, latency) pair. As described in Line 333~337, ProxylessNAS measures the latency of 5k full architectures to obtain the latency of each profiled layer following the paper[1] (roughly 42 unique blocks; 7 different operations per 6 different input shapes) following the original paper[1], which takes 5000*2s = 2.8h for GPU. Then we estimate the OFA’s building time of latency model in proportion to the number of unique blocks of OFA’s search space (roughly 225 blocks; 9 different operations per 25 different input shapes).
>
> [1] Cai, Han, Ligeng Zhu, and Song Han. "Proxylessnas: Direct neural architecture search on target task and hardware.", ICLR 2019.
>
> ***
> **(4) Check mathematical notations. f vs f^tau, V_h^0 : what is 0?**
>
> - $f^\tau$ (Line 125-127): $f$ refers to the meta-latency predictor meta-learned on the meta-training pool. $f^{\tau}$, on the other hand, refers to a **task-specific** latency predictor for task $\tau$, where the task $\tau$ is defined as a triplet of the dataset $X^{\tau}$, a set of latency measurements $Y^{\tau}$, and the device $h^\tau$.
>
> - $V_h^{(0)}$ (Line 156): We use the set of normalized latency values $V_h$ as a hardware embedding of a device, and $V_h^{(0)}$ denotes the set of original latency values before normalization.
>
> We will include the above descriptions in the revision for better clarity.
>
> ***
> **(5) Fig. 3 and 5: What is the unseen device? How to select the device on the pool for each point in the x-axis?**
>
> - The target unseen devices in Fig. 3, 5 are 6 meta-test devices listed in Line 228-232, which are Titan RTX GPU, Intel Xeon Gold CPU, Google Pixel2 mobile phone, Raspberry Pi 4, ASIC-Eyeriss, and FPGA. The plotted correlation values are average values across the 6 target devices. We will add this description in the revision for better clarity. In case of Fig. 5, devices of meta-training pool for the x-axis are randomly sampled as mentioned in Line 271 of the main paper.

---

> > ### Comment · Reviewer_rNFs · 2021-08-11
> > **Thank you for the answers**
> >
> > Thank you for the explanation. Please add the clarified details to the submitted paper.
> >
> > Just one more questions about the training details of HELP. In Section B of the supplementary file, there are four mains modules, do you link them as a single pipeline and train them together? It would be nice to clarify in the same section.

---

> > > ### Author Response · Authors · 2021-08-12
> > > **We appreciate the Reviewer rNFs’ response.**
> > >
> > > Yes. Our HELP is an end-to-end trainable model and thus all four modules are trained jointly. We will add this explanation in Section B and update the paper with all descriptions clarified by your valuable comments. We thank again the Reviewer rNFs’ time and effort in reviewing the paper.

---

### Official Review · Reviewer_TRAs · 2021-07-18

**Rating:** 7
**Confidence:** 2

**Summary:**

This work proposes a method which is able to learn a hardware performance predictor for a wide range of devices with only a few training samples(such as 10), through first one-time meta-learning a hardware embedding with the samples from a large device-architecture-performance pool and then adapt to the seen/unseen devices through inner-gradient manner with only a few training samples; throughout the process, this work treats the hardware devices as black boxes outputing the performance metric, thus not holding the statistical assumption about hardware devices' performance.



**Limitations And Societal Impact:**

Yes, it's addressed

**Main Review:**

Overall this work
1. Proposes a good method to predict the hardware performance of many devices while only requiring a few training samples without the statistical assumption between the hardware devices and their performance.
2.This work will certainly expedite and facilitate the process of hardware-aware NAS.
3.The writing is clear and to the points, pleasant to read

However, I have a couple of concerns which would be great if the authors can address them.

1. Algorithm-task level generality: It would be better to show some end-to-end results on more datasets/tasks, or model types(if possible), e.g., GNN, transformers...

2. More device variety and metric types: Different devices may have different calculation process for some performance metric, like throughput. For latency metric, most of the devices have a accumulative relation between layer(component)-wise latency and the system-latency. However, for throughput on architectures like cell-phones, Eyeriss or the FPGA in HW-NAS-Bench, the final throughput equals the inverse of system-latency, as they execute the workload layers sequentially. However, some architectures, such as the one in DNNBuilder,  execute the workload layers in a pipelined fashion. Thus the throughput will instead be the inverse of the bottleneck-layer latency. It would great to show some results/insights on how meta-learning on one type of devices generalize to the other type.

DNNBuilder: an Automated Tool for Building High-Performance DNN Hardware Accelerators for FPGAs

**Time Spent Reviewing:**

2 hours

---

> ### Author Response · Authors · 2021-08-10
> **Response to reviewer TRAs**
>
> We sincerely thank you for your constructive comments.
>
> **(1) Algorithm-task level generality: It would be better to show some end-to-end results on more datasets/tasks, or model types (if possible), e.g., GNN, transformers …**
>
> - We provided additional end-to-end NAS results on more devices in "Section C Additional Experiments" of the supplementary file, due to the page limit.  In Table C.4, we reported the results of latency-constrained NAS on two unseen platforms such as ASIC and FPGA, and on one unseen device (server CPU), on CIFAR-100 dataset with NAS-Bench-201 search space. In Table C.5, we provided the result of end-to-end NAS experiments on an unseen platform, including Edge GPU Jetson AGX Xavier, on CIFAR-100 dataset with the MobileNetV3 search space.
>
> - Further, we conducted more end-to-end latency constrained NAS experiments, on a new dataset **CIFAR10** with NAS-Bench-201 search space by combining HELP with MetaD2A as below Table. HELP is meta-trained on 18 devices, following the same setting from the experiments in the main paper, and is transferred to an unseen device.
>
> | Unseen Platform: FPGA 	|            	|              	|              	|                             	|
> |:---------------------:	|:----------:	|:------------:	|:------------:	|:---------------------------:	|
> |         Model         	| Const (ms) 	| Latency (ms) 	| Accuracy (%) 	| Samples From Target Devices 	|
> |    MetaD2A+BRP-NAS    	|      3     	|      2.7     	|     90.8     	|             900             	|
> |  **MetaD2A+HELP (Ours)**  	|      3     	|      2.4     	|     91.8     	|            **10**           	|
> |    MetaD2A+BRP-NAS    	|      4     	|      3.4     	|     92.7     	|             900             	|
> |  **MetaD2A+HELP (Ours)**  	|      4     	|      3.4     	|     93.1     	|            **10**           	|
> |    MetaD2A+BRP-NAS    	|      5     	|      6.8     	|     94.4     	|             900             	|
> |  **MetaD2A+HELP (Ours)**  	|      5     	|      4.7     	|     93.9     	|            **10**           	|
>
> | Unseen Device: Pixel2 	|            	|              	|              	|                             	|
> |:---------------------:	|:----------:	|:------------:	|:------------:	|:---------------------------:	|
> |         Model         	| Const (ms) 	| Latency (ms) 	| Accuracy (%) 	| Samples From Target Devices 	|
> |    MetaD2A+BRP-NAS    	|     14     	|      18      	|     92.9     	|             900             	|
> |  **MetaD2A+HELP (Ours)**  	|     14     	|      15      	|     92.6     	|            **10**           	|
> |    MetaD2A+BRP-NAS    	|     18     	|      23      	|     93.8     	|             900             	|
> |  **MetaD2A+HELP (Ours)**  	|     18     	|      19      	|     93.3     	|            **10**           	|
> |    MetaD2A+BRP-NAS    	|     22     	|      26      	|     94.4     	|             900             	|
> |  **MetaD2A+HELP (Ours)**  	|     22     	|      24      	|     93.7    	|            **10**           	|
>
> - We showed that HELP is successfully generalized on more end-to-end NAS tasks with 4 unseen devices in Section C of supplementary tile. In addition, we showed that HELP works well for an end-to-end NAS task on a new dataset, CIFAR10, in the tables above. We believe that this new experimental result will further strengthen our paper, and thank you for your helpful suggestion.
>
> - During the discussion period, we will provide more experiments to show the generality of HELP.
>
> ***
> **(2) More device variety and metric types: Different devices may have different calculation process for some performance metric, like throughput. For latency metric, most of the devices have an accumulative relation between layer(component)-wise latency and the system-latency. However, for throughput on architectures like cell-phones, Eyeriss or the FPGA in HW-NAS-Bench, the final throughput equals the inverse of system-latency, as they execute the workload layers sequentially. However, some architectures, such as the one in DNNBuilder, execute the workload layers in a pipelined fashion. Thus the throughput will instead be the inverse of the bottleneck-layer latency. It would be great to show some results/insights on how meta-learning on one type of device generalize to the other type.**
>
> - This is a very insightful question. Basically, the assumption of meta-learning is that the task distributions between meta-training and meta-test are similar. That is why transferring knowledge is helpful to meta-test tasks. As the reviewer said, if we tried to apply the meta-learned proposed approach on a target device that has a different calculation process on a metric with devices of the meta-training pool (e.g., DNNBuilder on throughput metric), meta-learning may be less effective, as the target device would be considered as an outlier of learned task distribution.
>
> - However, please note that the proposed hardware-adaptive modulator considers devices as a black-box function that outputs a metric value instead of directly modeling the metric calculation process or physical structure of hardware. Thus, regardless of a given device type, as long as the target device belongs to the task distribution, the knowledge would be helpful to rapidly adapt the predictor on the target device.
>
> - Also, further note that our **device-conditioned meta-learning** which is made possible by meta-learning the latency predictor conditioned on the given device embedding, and the hardware-aware modulator $z$ alleviates this issue, as shown in the ablation study and the comparison of our model to naive meta-learning baselines such as MAML, ANP, and Meta-SGD, in Table 2.
>
> ---
>
> We thank you for your insightful comments, and will include the above discussions and experiments in the revision.

---

> > ### Author Response · Authors · 2021-08-17
> > **Additional response to reviewer TRAs with end-to-end results for Transformers**
> >
> > **(1) Algorithm-task level generality: It would be better to show some end-to-end results on more datasets/tasks, or model types (if possible), e.g., GNN, transformers …**
> >
> > - During the discussion period, following the reviewer’s suggestion, we performed end-to-end latency-constraint NAS experiments for **Transformers** on machine translation task (WMT’14 En-De) to demonstrate algorithm-task level generality of the proposed approach (HELP). We plugged HELP to hardware-aware NAS method for Transformers (HAT) [1] and used data samples of GPU devices such as NVIDIA TitanX, NVIDIA 1080ti, NVIDIA 2080ti as meta-training pool and tested HAT + HELP on unseen devices such as NVIDIA TitanRTX and CPU Intel Xeon Gold6240. As a baseline model, we trained the end-to-end latency predictor (End-to-End Pred.) with 2000 samples for each device following HAT[1]. As shown below Table, HELP successfully searched for competitive Transformer models while using 200$\times$ fewer samples for training predictor on a machine translation task (WMT’14 En-De).
> > |               Target Device               	|          Model         	| Latency constraint 	| Real Latency 	| Samples From Target Device 	|  BLEU 	|
> > |:-----------------------------------------:	|:----------------------:	|:------------------:	|:------------:	|:--------------------------:	|:-----:	|
> > |    Unseen Device: NVIDIA GPU  TitanRTX    	| HAT + End-to-End Pred. 	|        90ms        	|     73.9     	|            2000            	| 27.08 	|
> > |                                           	|    HAT + HELP (Ours)   	|        90ms        	|     74.0     	|           **10**           	| 27.19 	|
> > |                                           	| HAT + End-to-End Pred. 	|        150ms       	|     108.4    	|            2000            	| 27.04 	|
> > |                                           	|    HAT + HELP (Ours)   	|        150ms       	|     106.5    	|           **10**           	| 27.44 	|
> > | Unseen Platform: CPU Intel Xeon  Gold6240 	| HAT + End-to-End Pred. 	|        200ms       	|     159.6    	|            2000            	| 27.20 	|
> > |                                           	|    HAT + HELP (Ours)   	|        200ms       	|     159.6    	|           **10**           	| 27.20 	|
> > |                                           	| HAT + End-to-End Pred. 	|        400ms       	|     369.4    	|            2000            	| 28.09 	|
> > |                                           	|    HAT + HELP (Ours)   	|        400ms       	|     343.2    	|           **10**           	| 27.52 	|
> >
> > - We will include these results in our manuscript. We believe these experiments strengthen our paper. Thank the reviewer for the constructive comments.
> >
> > [1] Wang, Hanrui, et al. "Hat: Hardware-aware transformers for efficient natural language processing." ACL (2020).

---

### Author Response · Authors · 2021-08-10
**Summary of the responses from all reviewers**

We sincerely appreciate your time and effort in reviewing our paper, as well as the positive feedbacks that the problem is well-motivated (RCcn) and important (FZ6V), the approach is convincing (rNFs), a great way (FZ6V), a nice application (RCcn), the contribution of the proposed work is outstanding (rNFs), the proposed work will certainly expedite and facilitate the process of hardware-aware NAS (TRAs).

We carefully read all your comments and have conducted multiple sets of experiments and provided answers to address some concerns from reviewers during the rebuttal period, as follows:

1. **FZ6V**: We showed the generalization performance of HELP on additional 18 unseen devices, which are used as the meta-training pool in the original paper.
***
2. **FZ6V** and **RCcn**: We included a correlation coefficient map among different devices within CPU/GPU/Mobile families to address the concern about the necessity of a hardware-adaptive meta-learning approach within the devices of the same platforms.
***
3. **TRAs**: We performed the more end-to-end latency-constrained NAS experiments on a new dataset to show the generalization performance of HELP.
***
4. **rNFs**: We clarified implementation and training details of hardware-adaptive modulator, mathematical expression, and experiment details.
***
5. **RCcn** and **FZ6V**: We revised the use of Big-O Notation of the proposed method.

We thank you for your constructive suggestions. We believe that the new discussions and experimental results to be highly valuable and strengthens our paper. We want to finally emphasize again that we tackle an important problem (Reducing the cost of the latency predictor for rapid hardware-aware NAS) and provide a novel method (Hardware-adaptive meta-learning framework) that is useful for many practitioners.

---

### Decision · Program_Chairs · 2021-09-27

**Decision:**

Accept (Spotlight)

**Comment:**

This is a timely paper. It addresses a relevant problem using novel meta-learning methods and obtains strong results.
All reviewers were in favour of acceptance.

There was a long internal discussion with dozens of posts about the comparison with BRP-NAS. The authors first misleadingly claimed that the results shown in the paper are consistent with the ones in the BRP-NAS paper, but they are clearly not (with a huge difference in Spearman correlation of roughly 0.8 vs. 0.99!). During the review process, the authors' code for running BRP-NAS was checked and found to be fine, so that is not the reason for the performance difference.
The most likely explanations for the different performance of BRP-NAS are (1) The data may be measured differently and/or (2) BRP-NAS may need different hyperparameters for new data. Concerning the latency measurement methodology, the BRP-NAS paper described their detailed methodology for measurements in detail: they pruned NAS-Bench-201 graphs and optimized them before running them on the device, then discarded the first few measurements and averaged multiple runs. The current paper doesn't give full details about the measurement pipeline to the paper, and I strongly encourage the authors to add more details. In the further discussion, the authors indeed explained that they use a less controlled (and arguably, more realistic) noise than in the BRP-NAS paper, which reduces the predictability.

The authors also added a convincing comparison to BRP-NAS on BRP-NAS' own benchmark LatBench.

An orthogonal concern that was brought up in the reviewer discussion was that Table 5 is overclaiming: OFA + HELP obviously does *not* take a total time of 26 seconds, as OFA takes very substantial time to construct the one-shot model. I strongly encourage the authors to fix this by reporting the total time spent, in order to present their contribution realistically.
Another concern that came up was that authors were given an additional page this year and encouraged to use this page to discuss the broader impact and societal impact; the authors instead deferred that discussion to the appendix, which could be seen as cheating the page limit. I strongly encourage them to include a discussion in the main paper for the final version and move some less important details to the appendix instead.

Overall, the work addresses an important and relevant problem; the use of meta-learning for NAS is much underexplored and thus very timely. I am therefore clearly in favour of acceptance.